# MetaMoE: Diversity-Aware Proxy Selection for Privacy-Preserving Mixture-of-Experts Unification

**Weisen Jiang** [1]   **Shuhao Chen** [2 3]   **Sinno Jialin Pan** [1]

## Abstract

Mixture-of-Experts (MoE) models scale capacity by combining specialized experts, but most existing approaches assume centralized access to training data. In practice, data are distributed across clients and cannot be shared due to privacy constraints, making unified MoE training challenging. We propose **MetaMoE**, a privacy-preserving framework that unifies independently trained, domain-specialized experts into a single MoE using public proxy data as surrogates for inaccessible private data. Central to Meta-MoE is diversity-aware proxy selection, which selects client-domain–relevant and diverse samples from public data to effectively approximate private data distributions and supervise router learning. These proxies are further used to align expert training, improving expert coordination at unification time, while a context-aware router enhances expert selection across heterogeneous inputs. Experiments on computer vision and natural language processing benchmarks demonstrate that MetaMoE consistently outperforms recent privacy-preserving MoE unification methods. Code is available at https://github.com/ws-jiang/MetaMoE.

## 1. Introduction

Large foundation models (Meta, 2024a;b; Qwen, 2024) have become indispensable across domains such as computer vision (CV) (Radford et al., 2021; Wei et al., 2024) and natural language processing (NLP) (Jiang et al., 2023; 2024; Chen et al., 2024; Lin et al., 2025; Jiang & Pan, 2025). In practice,

organizations and users often finetune a shared seed model on their own private data, resulting in a collection of specialized experts. While these experts could in principle be unified by aggregating private data to train a single Mixture-of-Experts (MoE) model (Jacobs et al., 1991; Fedus et al., 2022), data sharing is often infeasible due to confidentiality, regulatory, and ethical constraints. This raises a fundamental question (called Privacy-Preserving Mixture-of-Experts Unification): *How can we unify independently trained experts into one deployable MoE model while strictly preserving data privacy?* Although federated learning (Li et al., 2020; Kairouz et al., 2021; Zhang et al., 2021) enables collaborative training without data sharing, it requires costly synchronized optimization across many clients and often suffers from performance degradation under heterogeneous client data distributions (Wei et al., 2020).

Several approaches have been proposed to unify independently finetuned experts. BTM (Li et al., 2022) ensembles expert predictions, allowing embarrassingly parallel training but failing to produce a single deployable model for downstream fine-tuning or RLHF (Ouyang et al., 2022). Model averaging methods such as Model Soup (Wortsman et al., 2022) merge parameters directly, which is computationally efficient but fragile when experts are diverse. BTX (Sukhbaatar et al., 2024) extends this line by transplanting expert feed-forward network (FFN) sublayers into a shared MoE architecture with a learned router. However, its requirement for client-specific data to train this router limits its use in privacy-sensitive scenarios.

When private data cannot be shared, public data can be used as *proxies* to approximate unavailable private distributions and provide supervision for router learning. Most recently, FlexOlmo (Shi et al., 2025) adopts this strategy by training a router on proxy data while anchoring experts to a public model. However, its reliance on similarity-based proxy selection often produces redundant and narrowly concentrated proxies, limiting coverage of domain-relevant modes and weakening router supervision. Moreover, because experts are trained exclusively on private data and never exposed to proxies, they remain domain-isolated, further exacerbating the mismatch between expert behavior and proxy-based routing. These limitations motivate the need for a more

---

[1]Department of Computer Science and Engineering, The Chinese University of Hong Kong [2]Department of Computer Science and Engineering, Hong Kong University of Science and Technology [3]Department of Computer Science and Engineering, Southern University of Science and Technology. Correspondence to: Weisen Jiang <waysonkong@gmail.com>.

*Proceedings of the 43$^{rd}$ International Conference on Machine Learning*, Seoul, South Korea. PMLR 306, 2026. Copyright 2026 by the author(s).

principled proxy selection and alignment strategy for expert unification under privacy constraints.

We propose **MetaMoE**, a privacy-preserving framework for unifying independently trained experts into a single MoE model via *diversity-aware proxy selection*. Specifically, MetaMoE selects proxy samples using a relevance-weighted determinantal point process (DPP), an extension of standard DPPs (Macchi, 1975; Kulesza & Taskar, 2012) that augments diversity with client-specific relevance scores, ensuring that proxy samples are both diverse and representative of each client domain. Each client then performs proxy-aligned expert training by finetuning FFN sublayers on private data alongside proxy samples. Because the same proxy data are later used for router training, this exposure aligns expert behavior with the supervision available at unification time, mitigating domain isolation and enabling effective coordination across experts. A context-aware router that incorporates both token-level and sequence-level context further improves expert assignment, after which experts' FFN sublayers are merged into MoE layers and jointly finetuned on the union of proxy data. Experiments on both CV and NLP benchmarks demonstrate that MetaMoE consistently outperforms recent state-of-the-art baselines.

Our contributions are three-fold: (i) We study privacy-preserving MoE unification and propose MetaMoE, a framework that unifies independently finetuned experts without sharing private data, and provide formal privacy guarantees. (ii) We propose *diversity-aware proxy selection* via a relevance-weighted DPP, addressing the limitations of similarity-only proxy sampling for router learning. (iii) We design a proxy-aligned expert training strategy and a context-aware router, and demonstrate through extensive CV and NLP experiments that MetaMoE achieves superior performance over recent methods.

# 2. Related Works

## 2.1. Federated Learning

Federated Learning (FL) (Li et al., 2020; Kairouz et al., 2021; Zhang et al., 2021) enables collaborative training without centralizing raw data. Classical methods such as FedAvg (McMahan et al., 2017) aggregate local updates, with extensions for parameter-efficient tuning (Hu et al., 2022) or differential privacy (Dwork & Roth, 2014). Yet scaling FL to large models (Meta, 2024a;b; Radford et al., 2021) is difficult due to costly synchronization, degraded generalization (Zhang et al., 2023), and privacy leakage from gradients (Wang et al., 2020; Darzi et al., 2024). In contrast, MetaMoE trains experts independently and asynchronously, then merges them via proxy-data-driven routing, avoiding FL's communication bottlenecks while preserving privacy.

## 2.2. Model Merging and Mixture-of-Experts

**Model Merging** (Yang et al., 2024; Wortsman et al., 2022; Ilharco et al., 2023; Yadav et al., 2023; Rame et al., 2023; Li et al., 2024) explores how to combine multiple independently trained models into a single, stronger model without requiring costly joint training. Branch-Train-Merge (BTM) (Li et al., 2022) trains domain experts independently and ensembles outputs at inference time, but does not yield a single unified model (hindering downstream SFT/RLHF (Ouyang et al., 2022) and incurring inference overhead). Model Soup (Wortsman et al., 2022) shows that averaging the weights of finetuned models often improves both accuracy and robustness with no additional inference cost, yet it is fragile when experts diverge in function space, leading to degraded performance in heterogeneous settings. Branch-Train-MiX (BTX) (Sukhbaatar et al., 2024) inserts experts into MoE layers and learns a post-hoc router via additional fine-tuning on private data. More recently, Flex-Olmo (Shi et al., 2025) advances this direction in federated settings by anchoring experts to a shared public model and aligning them with router embeddings, where proxy samples are selected for each client based solely on similarity. In contrast, our MetaMoE leverages a relevance-and-diversity criterion via a relevance-weighted DPP to select proxies, and employs a context-aware router, thereby combining heterogeneous experts into a unified MoE model under privacy constraints.

The **Mixture-of-Experts (MoE)** (Jacobs et al., 1991) framework enhances model flexibility by combining specialized experts via a gating mechanism. Modern MoE architectures (Riquelme et al., 2021; Fedus et al., 2022) scale Transformers by activating only a sparse subset of experts per token, expanding capacity without proportional compute cost (Shazeer et al., 2017). Key designs include the Switch Transformer (Fedus et al., 2022) with top-1 routing and variants (Shazeer et al., 2017; Riquelme et al., 2021; Roller et al., 2021; Dai et al., 2024) exploring top-$k$ gating, stochastic routing, and random assignment for better accuracy, efficiency, and load balance. While MoE is effective for scaling, these methods rely on centralized access to all training data and synchronized training. To overcome these limits under privacy constraints, we propose MetaMoE, which introduces proxy-data-driven routers and decentralized expert training.

## 2.3. Determinantal Point Processes (DPPs)

Determinantal Point Processes (DPPs) (Macchi, 1975; Kulesza & Taskar, 2012; Lavancier et al., 2015) are probabilistic models designed to capture *negative interactions* among items, which are useful for diverse subset selection. Formally, consider a discrete set $\mathcal{Z} = \{1, 2, \ldots, N\}$, where each element corresponds to an item with feature vector $\mathbf{x}_i$. A DPP defines a probability distribution over all $2^N$

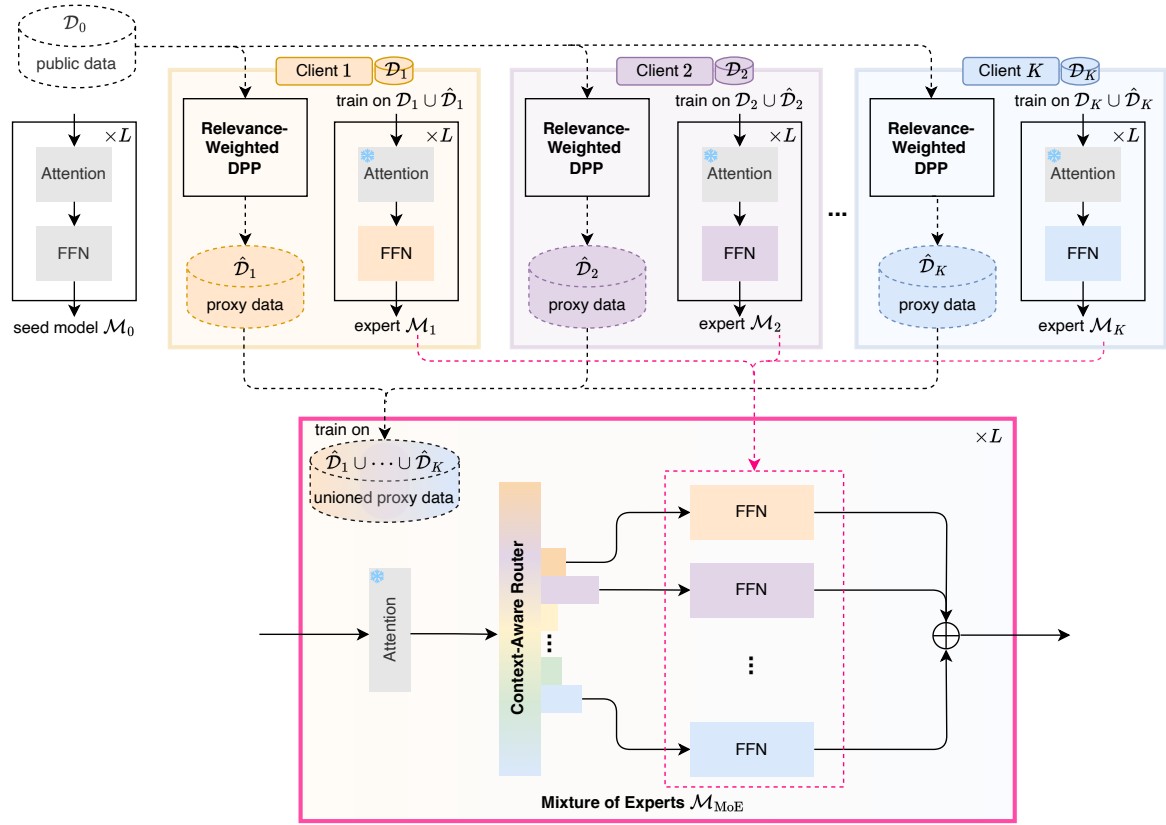

*Figure 1.* Illustration of MetaMoE.

subsets of $\mathcal{Z}$. To specify the distribution, we construct a positive semi-definite kernel matrix $\mathbf{L} \in \mathbb{R}^{N \times N}$, $\mathbf{L}_{ij} = \kappa(\mathbf{x}_i, \mathbf{x}_j)$, where $\kappa(\cdot, \cdot)$ is a kernel function encoding similarity. The probability of sampling a subset $\mathcal{S} \subseteq \mathcal{Z}$ is:

$$\mathbb{P}(\mathcal{S}) = \frac{\det(\mathbf{L}_\mathcal{S})}{\det(\mathbf{L} + \mathbf{I})}, \tag{1}$$

where $\mathbf{L}_\mathcal{S}$ is the submatrix indexed by $\mathcal{S}$, $\det(\cdot)$ is the determinant, and $\mathbf{I}$ is the identity matrix. The denominator $\det(\mathbf{L} + \mathbf{I})$ is constant with respect to the choice of $\mathcal{S}$, so for subset selection it can be ignored; maximizing the selection probability $\mathbb{P}(\mathcal{S})$ is thus equivalent to maximizing $\det(\mathbf{L}_\mathcal{S})$.

By reproducing kernel Hilbert space representation (Schölkopf et al., 2001), one may write $\kappa(\mathbf{x}_i, \mathbf{x}_j) = \phi(\mathbf{x}_i)^\top \phi(\mathbf{x}_j)$ for some feature map $\phi(\cdot)$. Then $\det(\mathbf{L}_\mathcal{S})$ equals the squared volume of a parallelotope spanned by $\{\phi(\mathbf{x}_i) \mid i \in \mathcal{S}\}$, so similar items are less likely to be selected together, promoting diversity.

## 3. Methodology

### 3.1. Problem Formulation

Denote by $\mathcal{M}_0$ a seed model and by $\mathcal{D}_0$ a publicly available dataset. We consider $K$ clients, where each client $p$

owns a private dataset $\mathcal{D}_p$ from its local domain. Sharing $\{\mathcal{D}_p\}_{p=1}^K$ directly is prohibited due to privacy constraints. Each client adapts the seed model locally to obtain a domain-specialized expert $\mathcal{M}_p$. Our objective is to unify these experts $\{\mathcal{M}_p\}_{p=1}^K$ into a Mixture-of-Experts (MoE) model $\mathcal{M}_{\mathrm{MoE}}$ that can be deployed back to all clients such that each client can achieve multi-domain capabilities. In the MoE, experts encode domain-specific knowledge, while the router coordinates the experts to enable effective collaboration.

*The core challenge lies in training the router.* Conventional MoE training (Jacobs et al., 1991; Shazeer et al., 2017) assumes centralized access to all client data, but in our setting only the public dataset $\mathcal{D}_0$ is globally accessible. Thus, the router must be learned without directly observing $\{\mathcal{D}_p\}_{p=1}^K$, while still generalizing across all clients' domains. We address this challenge by selecting proxy samples from $\mathcal{D}_0$ to approximate each $\mathcal{D}_p$, enabling the router to coordinate domain-specific experts in a privacy-preserving manner. Figure 1 illustrates our proposed MetaMoE, consisting of three stages (proxy data selection, proxy-aligned expert training, and context-aware router training) and will be detailed in the following sections.

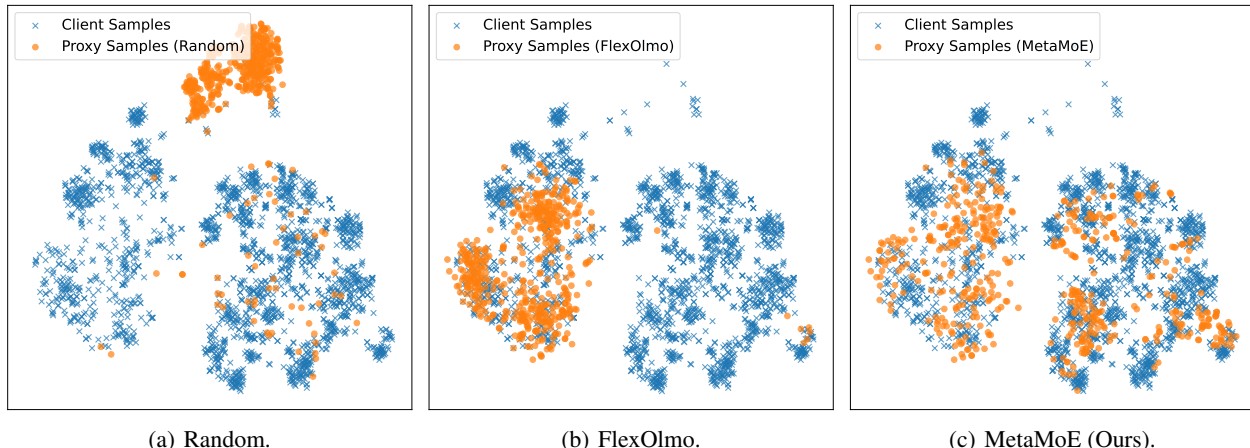

(a) Random.  (b) FlexOlmo.  (c) MetaMoE (Ours).

*Figure 2.* t-SNE visualization of selected proxy samples with random selection, FlexOlmo selection (relevance only), and our selection (relevance + diversity) for Pets with ViT-B/32 as the seed model. As can be seen, our selection yields a more diverse and representative proxy dataset that covers the private-data manifold more effectively (see Section 4.3 for further analysis).

## 3.2. Proxy Data Selection via Relevance-Weighted DPP

Since client private data is inaccessible for router training, we construct a proxy dataset $\hat{\mathcal{D}}_p$ for each client $p$ from the public dataset $\mathcal{D}_0$ to serve as a proxy for $\mathcal{D}_p$. Effective proxy data should satisfy two criteria: they should be both **relevant** to the private data $\mathcal{D}_p$ and sufficiently **diverse** to avoid redundancy. Relevance ensures that the proxy samples resemble the private data so that the router trained on proxies learns domain-appropriate decision boundaries, rather than being distracted by unrelated public samples. Diversity, on the other hand, ensures that the selected proxy samples cover different regions of the private-data manifold, rather than clustering around a narrow region of highly similar samples, thereby providing broader coverage and improving the router's generalization.

Determinantal point processes (DPPs) (Macchi, 1975; Kulesza & Taskar, 2012; Lavancier et al., 2015) naturally enforce diversity, but a vanilla DPP (Section 2.3) ignores whether the chosen samples align with the client's domain $\mathcal{D}_p$. Hence, naively applying a vanilla DPP may select a diverse yet irrelevant proxy dataset. To overcome this, we propose a *relevance-weighted DPP*, which augments the kernel with client-specific relevance scores:

$$\tilde{\kappa}(\mathbf{x}_i, \mathbf{x}_j) = g(\mathbf{x}_i, \mathcal{D}_p)\, \kappa(\mathbf{x}_i, \mathbf{x}_j)\, g(\mathbf{x}_j, \mathcal{D}_p), \quad (2)$$

where $\kappa(\mathbf{x}_i, \mathbf{x}_j)$ measures similarity between public samples (e.g., cosine similarity), and $g(\mathbf{x}_i, \mathcal{D}_p)$ quantifies the relevance of $\mathbf{x}_i$ to $\mathcal{D}_p$ (e.g., via a classifier distinguishing $\mathcal{D}_0$ from $\mathcal{D}_p$, see Appendix A). This yields the *relevance-weighted kernel matrix*

$$\widetilde{\mathbf{L}} = \mathrm{Diag}(\mathbf{r})\, \mathbf{L}\, \mathrm{Diag}(\mathbf{r}), \quad (3)$$

where $\mathbf{L}_{ij} = \kappa(\mathbf{x}_i, \mathbf{x}_j)$, $\mathbf{r} = [g(\mathbf{x}_1, \mathcal{D}_p), \dots, g(\mathbf{x}_N, \mathcal{D}_p)]$, and $\mathrm{Diag}(\mathbf{r})$ denotes the diagonal matrix with $\mathbf{r}$ on the di-

agonal. According to (1) and (3), the unnormalized log-probability of selecting a subset $\mathcal{S}$ under relevance-weighted DPP is given by

$$\log \det(\widetilde{\mathbf{L}}_{\mathcal{S}}) = 2 \sum_{i \in \mathcal{S}} \log \mathbf{r}_i \;+\; \log \det(\mathbf{L}_{\mathcal{S}}), \quad (4)$$

where the first term $2 \sum_{i \in \mathcal{S}} \log \mathbf{r}_i$ encourages relevance by favoring samples closer to the client's data $\mathcal{D}_p$, while the second term $\log \det(\mathbf{L}_{\mathcal{S}})$ is the standard DPP repulsion term that enforces diversity.

The proxy dataset $\hat{\mathcal{D}}_p$ is selected by

$$\hat{\mathcal{D}}_p = \underset{\mathcal{S} \subseteq \mathcal{Z}, |\mathcal{S}| = m}{\arg\max} \; \log \det(\widetilde{\mathbf{L}}_{\mathcal{S}}), \quad (5)$$

which yields a *relevance-weighted and diverse* cover of the private-data manifold. In contrast to FlexOlmo's relevance-only proxy selection (Shi et al., 2025), which often collapses onto redundant samples and provides a narrow view of a client's domain (see Figure 2(b)), the relevance-weighted DPP explicitly balances relevance and diversity. The diversity term discourages near-duplicate proxies, providing the router with a richer supervision signal that better spans the private-domain manifold (see Figure 2(c)) and enables more effective expert coordination. Because all proxies are selected from a public dataset, choosing client-specific proxy subsets does not violate privacy constraints (see Section 3.6 and Appendix E.4).

Since exact maximum a posteriori (MAP) inference in (5) is NP-hard, we adopt efficient greedy algorithms with approximation guarantees (Kulesza & Taskar, 2012; Gillenwater et al., 2012; Han et al., 2017). We first restrict a candidate pool (the top-$n$ public samples ranked by $g(\mathbf{x}_i, \mathcal{D}_p)$), and then perform greedy MAP inference to construct $\hat{\mathcal{D}}_p$ by iteratively adding the sample that maximizes the marginal gain in $\log \det(\widetilde{\mathbf{L}}_{\hat{\mathcal{D}}_p})$. Using Cholesky updates (Horn & Johnson,

1985), the computational cost is reduced from $\mathcal{O}(nm^3)$ to $\mathcal{O}(nm)$ (see Appendix B), where $n$ is the candidate pool size and $m$ is the target proxy set size.

## 3.3. Proxy-Aligned Expert Training

For each client $p$, an expert is initialized by branching from the shared seed model $\mathcal{M}_0$. Only the feed-forward network (FFN) sublayers are finetuned using a combination of private data $\mathcal{D}_p$ and client-specific proxy data $\hat{\mathcal{D}}_p$, while all other parameters remain frozen.

Training experts solely on private data, as in (Shi et al., 2025), produces highly specialized models but leads to *domain isolation*: since the router is trained later without access to private data, it struggles to coordinate experts adapted to heterogeneous domains.

We address this mismatch by incorporating proxy data during expert training. Since router learning relies exclusively on proxy data, exposing each expert to its corresponding proxies calibrates expert representations to the same data distribution used for routing. Specifically, the router is trained on the union $\hat{\mathcal{D}}_1 \cup \cdots \cup \hat{\mathcal{D}}_K$; aligning expert training with this supervision improves routing compatibility while preserving data privacy. Compared with FlexOlmo, which trains experts only on private data, proxy-aligned training in MetaMoE preserves domain-specific expertise while enabling more effective expert coordination.

## 3.4. Context-Aware Router for Expert Unification

After collecting domain-specific experts $\{\mathcal{M}_p\}_{p=1}^{K}$, we merge their FFN sublayers into MoE modules at each Transformer layer. We denote $\text{FFN}_p^{(l)}(\cdot)$ as the $l$-th FFN sublayer of the $p$-th expert $\mathcal{M}_p$ and let $\mathbf{z}_t^{(l)}$ be the $t$-th token representation of input sequence $\mathbf{x}$ in the $l$-th layer. The corresponding MoE module is then formulated as

$$\mathcal{M}_{\text{MoE}}^{(l)}(\mathbf{z}_t^{(l)}) = \sum_{p \in \text{Top-}k(\pi^{(l)}(\mathbf{z}_t^{(l)}))} [\pi^{(l)}(\mathbf{z}_t^{(l)})]_p \cdot \text{FFN}_p^{(l)}(\mathbf{z}_t^{(l)}), \quad (6)$$

where $\pi^{(l)}(\mathbf{z}_t)$ is the router's score distribution over experts, and Top-$k(\cdot)$ is the top-$k$ selection.

Conventional routers rely solely on $\mathbf{z}_t^{(l)}$, making routing decisions based on token-level features. However, this can be unreliable: tokens with similar surface forms may belong to different domains and require different experts. Such routing collisions are particularly problematic here, since the router is trained only on proxies and never directly observes the true client data distributions.

To mitigate this, we introduce a context-aware router. Instead of routing purely from $\mathbf{z}_t^{(l)}$, we form a blended representation $\tilde{\mathbf{z}}_t^{(l)} = (1 - \lambda)\,\mathbf{z}_t^{(l)} + \lambda\,\mathbf{z}_{\mathbf{x}}^{(l)}$, where $\mathbf{z}_{\mathbf{x}}^{(l)} =$

$\frac{1}{T} \sum_{t=1}^{T} \mathbf{z}_t^{(l)}$ is a sequence-level embedding capturing global context ($T$ is the length of $\mathbf{x}$), and $\lambda \in [0, 1]$ is a learnable weight. This blending balances token semantics with broader context cues, and routing distribution is computed as

$$\pi^{(l)}(\mathbf{z}_t^{(l)}) = \text{softmax}[\tilde{\mathbf{z}}_t^{(l)\top} \mathbf{e}_1^{(l)}, \ \ldots, \ \tilde{\mathbf{z}}_t^{(l)\top} \mathbf{e}_K^{(l)}], \quad (7)$$

where $\mathbf{e}_p^{(l)}$ is the learnable routing vector for expert $p$, initialized as the mean embedding of $\mathcal{D}_p \cup \hat{\mathcal{D}}_p$, i.e., $\mathbf{e}_p^{(l)} = \frac{1}{|\mathcal{D}_p \cup \hat{\mathcal{D}}_p|} \sum_{\mathbf{x} \in \mathcal{D}_p \cup \hat{\mathcal{D}}_p} \mathcal{M}_p^{(1:l)}(\mathbf{x})$, with $\mathcal{M}_p^{(1:l)}(\cdot)$ denoting the first $l$ layers of $\mathcal{M}_p$. This domain-aware initialization injects each expert's domain characteristics directly into the router, giving it meaningful expert–token priors.

## 3.5. Final MoE Training

At the final stage, we aggregate all proxy datasets $\hat{\mathcal{D}}_1 \cup \cdots \cup \hat{\mathcal{D}}_K$ and finetune the unified MoE model. This process updates the router while jointly adapting the FFN experts under supervision from proxy data, ensuring that experts are not treated as isolated components but instead operate cohesively within an MoE architecture. Hence, the resulting model $\mathcal{M}_{\text{MoE}}$ unifies domain-specific expertise with a privacy-preserving router, yielding a unified MoE model that generalizes across heterogeneous client domains. The overall procedure of MetaMoE is summarized in Algorithm 1.

## 3.6. Privacy Analysis

We specify the threat model and provide formal privacy guarantees for the artifacts communicated by MetaMoE.

Throughout the MetaMoE pipeline, each client $p$ communicates only the following artifacts to the central server: (i) indices of selected public proxy samples (a subset of $\mathcal{D}_0$), (ii) final expert weights (FFN sublayers of $\mathcal{M}_p$), and (iii) routing vectors $\mathbf{e}_p^{(l)}$ (mean embeddings used to initialize the router, (7)). Raw private samples, gradients, and intermediate activations are never shared. This privacy model, commonly referred to as *data residency* (McMahan et al., 2017), is standard in privacy-preserving collaborative learning. Notably, MetaMoE is strictly more conservative than federated learning (FL) (Li et al., 2020; Kairouz et al., 2021; Zhang et al., 2021): it transmits only final expert weights in a single one-shot communication, whereas FL iteratively exchanges gradient updates that are susceptible to model inversion attacks (Wang et al., 2020).

Among the three communicated artifacts, proxy indices are deterministic functions of public data and carry no private information. Expert weights, as outputs of deep network training, are shared in the same manner as in FL. The routing vectors $\mathbf{e}_p^{(l)}$, while computed over both private and proxy data, also carry negligible private information, as we for-

---

**Algorithm 1** MetaMoE.

---

**Require:** public dataset $\mathcal{D}_0$; client private datasets $\{\mathcal{D}_p\}_{p=1}^K$; kernel $\kappa(\cdot, \cdot)$; proxy dataset size $m$; optional candidate pool size $n$ ($m \leq n \ll |\mathcal{D}_0|$);

1: **for** each client $p = 1, 2, \ldots, K$ **do**
2:      // select proxy samples by relevance-weighted DPP
3:      Train a binary classifier $g(\mathbf{x}, \mathcal{D}_p)$ to distinguish $\mathcal{D}_0$ vs. $\mathcal{D}_p$;
4:      Compute relevance $r(\mathbf{x}) = g(\mathbf{x}, \mathcal{D}_p)$ for each $\mathbf{x} \in \mathcal{D}_0$;
5:      Let $\mathcal{C}_p \subseteq \mathcal{D}_0$ be the top-$n$ elements of $\mathcal{D}_0$ ordered by $r(\mathbf{x})$;
6:      Construct kernel matrix $\mathbf{L} \in \mathbb{R}^{n \times n}$ with $\mathbf{L}_{ij} = \kappa(\mathbf{x}_i, \mathbf{x}_j)$ for each $\mathbf{x}_i, \mathbf{x}_j \in \mathcal{C}_p$;
7:      Form $\mathbf{r} = [\, r(\mathbf{x}) \,]_{\mathbf{x} \in \mathcal{C}_p}$ and relevance-weighted kernel matrix $\widetilde{\mathbf{L}} = \mathrm{Diag}(\mathbf{r})\, \mathbf{L}\, \mathrm{Diag}(\mathbf{r})$;
8:      Greedily build $\hat{\mathcal{D}}_p$ of size $m$ by iteratively adding $\mathbf{x}^\star = \underset{\mathbf{x} \in \mathcal{C}_p \setminus \hat{\mathcal{D}}_p}{\arg\max}\ \log \det(\tilde{\mathbf{L}}_{\hat{\mathcal{D}}_p \cup \{\mathbf{x}\}}) - \log \det(\tilde{\mathbf{L}}_{\hat{\mathcal{D}}_p})$;
9:      // proxy-aligned expert training
10:      Finetune the FFN sublayers of the $p$-th client model on $\hat{\mathcal{D}}_p \cup \mathcal{D}_p$, and freeze all other sublayers;
11:      For each layer $l$, compute router vector $\mathbf{e}_p^{(l)} = \frac{1}{|\hat{\mathcal{D}}_p \cup \mathcal{D}_p|} \sum_{\mathbf{x} \in \hat{\mathcal{D}}_p \cup \mathcal{D}_p} \mathcal{M}_p^{(1:l)}(\mathbf{x})$;
12: **end for**
13: // build the MoE model with context-aware router
14: Merge all clients' FFN sublayers into a single MoE model $\mathcal{M}_{\mathrm{MoE}}$;
15: Collect $\{\mathbf{e}_p^{(l)}\}_{p=1}^K$ for all layers $l = 1, \ldots, L$ to initialize the router;
16: Compute the routing distribution $\pi^{(l)}(\mathbf{z}_t^{(l)})$ for each layer $l$ using (7);
17: Finetune $\mathcal{M}_{\mathrm{MoE}}$ on the union of proxy datasets $\hat{\mathcal{D}}_1 \cup \cdots \cup \hat{\mathcal{D}}_K$;
18: **return** Trained MoE model $\mathcal{M}_{\mathrm{MoE}}$.

---

mally establish below.

We analyze the routing vector for a fixed client $p$ and layer $l$. Let $N = |\mathcal{D}_p|$ and $m = |\hat{\mathcal{D}}_p|$ denote the private and proxy dataset sizes, respectively, and let $f(\cdot) = \mathcal{M}_p^{(1:l)}(\cdot)$ denote the first $l$ layers of $\mathcal{M}_p$ used as the encoder. Define the mean private embedding $\boldsymbol{\mu}_{\mathrm{priv}} = \frac{1}{N} \sum_{\mathbf{x} \in \mathcal{D}_p} f(\mathbf{x})$ and the mean proxy embedding $\boldsymbol{\mu}_{\mathrm{proxy}} = \frac{1}{m} \sum_{\mathbf{x} \in \hat{\mathcal{D}}_p} f(\mathbf{x})$. The routing vector (7) can be decomposed as

$$\mathbf{e}_p^{(l)} = \frac{N}{N+m}\, \boldsymbol{\mu}_{\mathrm{priv}} + \frac{m}{N+m}\, \boldsymbol{\mu}_{\mathrm{proxy}}. \tag{8}$$

An adversary observes $\mathbf{e}_p^{(l)}$ and knows $\boldsymbol{\mu}_{\mathrm{proxy}}$, $m$, and $f$ (all derived from public information), but **does not know** $N$ (the private dataset size is never communicated). We establish three formal guarantees (full derivations in Appendix F):

*(i) Per-sample sensitivity is bounded by $O(1/m)$.* Let $B = \max_{\mathbf{x}} \|f(\mathbf{x})\|_2$ be the embedding norm bound. Replacing any single private sample $\mathbf{x}_k \in \mathcal{D}_p$ with an arbitrary $\mathbf{x}_k'$ yields a perturbed routing vector $\mathbf{e}_p'^{(l)}$ satisfying

$$\|\mathbf{e}_p^{(l)} - \mathbf{e}_p'^{(l)}\|_2 = \frac{\|f(\mathbf{x}_k') - f(\mathbf{x}_k)\|_2}{N+m} \leq \frac{2B}{N+m} \leq \frac{2B}{m}, \quad (9)$$

where the last inequality uses $N + m \geq m$. Thus the proxy set size $m$ alone provides a universal sensitivity ceiling, independent of both $N$ and the domain gap.

*(ii) The mean private embedding $\boldsymbol{\mu}_{\mathrm{priv}}$ is unrecoverable.* From (8), isolating $\boldsymbol{\mu}_{\mathrm{priv}}$ requires computing $\boldsymbol{\mu}_{\mathrm{priv}} = \big((N +$

$m)\, \mathbf{e}_p^{(l)} - m\, \boldsymbol{\mu}_{\mathrm{proxy}}\big)/N$, which cannot be evaluated without knowing $N$. Since $\boldsymbol{\mu}_{\mathrm{priv}}$ is the coarsest possible summary of private data (a single vector), finer-grained distributional properties (e.g., variance, class proportions, individual samples) are a fortiori unrecoverable from $\mathbf{e}_p^{(l)}$.

*(iii) MetaMoE exposes strictly less private information than FlexOlmo.* FlexOlmo (Shi et al., 2025) shares per-expert routing embeddings computed as mean embeddings over private data alone (Section 3.3.2 of the FlexOlmo paper), directly exposing $\boldsymbol{\mu}_{\mathrm{priv}}$. In contrast, MetaMoE dilutes the private component by averaging over both private and proxy data (8), providing a strictly stronger privacy guarantee.

## 4. Experiments

### 4.1. Experiments on CV Tasks

**Datasets.** We evaluate on three benchmarks from distinct domains: (i) Pets (Parkhi et al., 2012), with 37 cat and dog breeds, (ii) Flowers (Nilsback & Zisserman, 2008), with 102 flower categories, and (iii) EuroSAT (Helber et al., 2019), with 10 land-use classes from satellite imagery. Each dataset serves as a client domain, covering fine-grained categorization, natural object recognition, and remote sensing. As the public dataset $\mathcal{D}_0$, we adopt ImageNet (Deng et al., 2009), which is a large-scale visual corpus providing 1.28M images across 1,000 categories.

**Models.** We adopt CLIP ViT-B/32 and ViT-B/16 (Rad-

ford et al., 2021) as the seed models, which consist of a transformer-based image encoder and a text encoder.

**Implementation Details.** For each client, we build a proxy dataset by first selecting a candidate pool of $n = 3000$ ImageNet samples that are most similar to its private data according to the relevance score $g(\mathbf{x}, \mathcal{D}_p)$, then choosing $m = 500$ proxy samples via relevance-weighted DPP with the cosine similarity kernel $\kappa(\mathbf{x}_i, \mathbf{x}_j) = \cos(\mathbf{z}_i, \mathbf{z}_j)$, where $\mathbf{z}_i$ is $\mathbf{x}_i$'s embedding extracted from $\mathcal{M}_0$. Client models are initialized from the seed model $\mathcal{M}_0$ and finetuned on both private and proxy data. For proxy-aligned expert training, we finetune the FFN sublayers of the visual encoder using LoRA (Hu et al., 2022) (rank 16, scaling factor $\alpha = 32$) for 10 epochs with the SGD optimizer (momentum 0.9, weight decay 0.0001, learning rate 0.01, batch size 128, constant learning rates schedule). For router training, we adopt top-1 routing and finetune for 5 epochs using SGD (lr=0.001).

**Baselines.** We compare our method against various baselines: (i) ZeroShot directly evaluates the seed model $\mathcal{M}_0$ without any adaptation, providing a capacity-only lower bound. (ii) BTM (Li et al., 2022) ensembles predictions from independently trained experts at inference, but does not produce a unified model for downstream use. (iii) ModelSoup (Wortsman et al., 2022) merges experts by averaging their weights into a single model, avoiding inference overhead from ensembling but losing specialization when experts diverge. (iv) BTX (Sukhbaatar et al., 2024) integrates experts by inserting their FFN sublayers into MoE layers, averaging the remaining parameters, and then fine–tuning the mixed model on public data (since private data are unavailable), following the setup of FlexOlmo (Shi et al., 2025). (v) FlexOlmo (Shi et al., 2025) aligns experts without centralizing private data by anchoring them to a shared public model and introducing per-expert router embeddings, which are later finetuned on proxy data selected by similarity to private domains. (vi) Separate Experts evaluates each independently finetuned expert across all domains. (vii) UnrestrictedMoE trains a unified MoE model directly on the merged private datasets from all clients, thereby achieving strong performance but breaking privacy constraints.

**Results.** Tables 1 and 2 report the testing accuracy on the three client domains when using CLIP ViT-B/32 and CLIP ViT-B/16 as the seed model, respectively. Across both backbones, our proposed MetaMoE consistently outperforms all privacy-preserving baselines. Compared with BTM, which ensembles predictions from independent experts without parameter sharing, MetaMoE integrates experts at the parameter level and leverages a router to coordinate experts, thereby achieving better performance. Compared with ModelSoup—which averages model parameters—MetaMoE achieves much higher accuracy by explicitly retaining expert specialization within a unified MoE

architecture. In contrast to BTX and FlexOlmo, which also employ MoE-style unification but rely on proxy data without explicit distribution alignment, MetaMoE yields clear gains by combining relevance-weighted DPP-selected proxy subsets with proxy-aligned expert training, allowing routers to coordinate experts more effectively. Quantitatively, MetaMoE achieves an average accuracy of 94.52% with CLIP ViT-B/32 and 96.24% with CLIP ViT-B/16, outperforming the strongest baseline FlexOlmo (92.92% and 93.53%) by about 1.6 and 2.7 points, respectively.

*Table 1.* Accuracy of CV Tasks when using CLIP ViT-B/32 as the seed model.

|  | Pets | Flowers | EuroSAT | **Average** |
|---|---|---|---|---|
| UnrestrictedMoE | 92.45 | 96.43 | 98.15 | 95.68 |
| ZeroShot | 85.77 | 61.59 | 29.81 | 59.06 |
| Expert I (Pets) | 92.40 | 59.03 | 22.74 | 58.06 |
| Expert II (Flowers) | 84.25 | 96.91 | 27.21 | 69.46 |
| Expert III (EuroSAT) | 82.64 | 52.42 | 97.91 | 77.66 |
| BTM | 90.81 | 85.10 | 95.07 | 90.33 |
| ModelSoup | 87.90 | 70.52 | 64.19 | 74.20 |
| BTX | 88.44 | 75.07 | 59.38 | 74.30 |
| FlexOlmo | 91.36 | 90.62 | 96.79 | 92.92 |
| MetaMoE | **91.91** | **93.67** | **97.98** | **94.52** |

*Table 2.* Testing Accuracy of CV Tasks when using CLIP ViT-B/16 as the seed model.

|  | Pets | Flowers | EuroSAT | **Average** |
|---|---|---|---|---|
| UnrestrictedMoE | 94.30 | 97.73 | 98.23 | 96.75 |
| ZeroShot | 88.53 | 68.09 | 34.00 | 63.54 |
| Expert I (Pets) | 94.44 | 65.69 | 30.10 | 63.41 |
| Expert II (Flowers) | 85.99 | 98.13 | 24.05 | 69.39 |
| Expert III (EuroSAT) | 86.56 | 60.05 | 98.43 | 81.68 |
| BTM | 93.21 | 84.65 | 97.38 | 91.75 |
| ModelSoup | 89.94 | 74.18 | 74.14 | 79.42 |
| BTX | 89.89 | 78.52 | 75.20 | 81.20 |
| FlexOlmo | 94.09 | 89.40 | 97.11 | 93.53 |
| MetaMoE | **94.22** | **97.08** | **97.41** | **96.24** |

### 4.2. Experiments on NLP Tasks

**Datasets.** We evaluate on three benchmarks for commonsense reasoning: (i) CommonsenseQA (Talmor et al., 2019) (denoted as CSQA), which tests general world knowledge through discrimination among semantically related concepts, (ii) CosmosQA (Huang et al., 2019), which emphasizes narrative reasoning by requiring inference of implicit causes and effects, and (iii) SocialIQA (Sap et al., 2019), which focuses on social commonsense involving human actions, motivations, and social implications. Proxy data are selected from Alpaca (Taori et al., 2023), a publicly available collection of 52K instruction–response pairs that spans diverse instruction-tuning tasks. Note that the public dataset (Alpaca) has no domain overlap with the client datasets.

**Models.** We evaluate on two LLaMA models across dif-

*Table 3.* Accuracy of NLP Tasks when using LLaMA-3.2-3B as the seed model.

| | CSQA | CosmosQA | SocialIQA | **Average** |
|---|---|---|---|---|
| UnrestrictedMoE | 75.51 | 78.39 | 71.80 | 75.23 |
| ZeroShot | 62.49 | 62.68 | 56.19 | 60.45 |
| Expert I (CSQA) | 74.94 | 70.18 | 60.54 | 68.55 |
| Expert II (CosmosQA) | 63.06 | 78.69 | 58.96 | 66.90 |
| Expert III (SocialIQA) | 65.44 | 68.04 | 72.42 | 68.63 |
| BTM | 74.61 | 75.44 | 68.17 | 72.74 |
| ModelSoup | 73.71 | 75.24 | 71.75 | 73.57 |
| BTX | 69.62 | 72.06 | 71.75 | 71.14 |
| FlexOlmo | 73.30 | 73.33 | 70.88 | 72.50 |
| MetaMoE | **74.94** | **76.05** | **72.26** | **74.42** |

*Table 4.* Accuracy of NLP Tasks when using LLaMA-3.1-8B as the seed model.

| | CSQA | CosmosQA | SocialIQA | **Average** |
|---|---|---|---|---|
| UnrestrictedMoE | 81.90 | 85.70 | 76.41 | 81.34 |
| ZeroShot | 69.37 | 75.98 | 57.57 | 67.64 |
| Expert I (CSQA) | 81.24 | 77.29 | 69.19 | 75.91 |
| Expert II (CosmosQA) | 71.42 | 86.23 | 67.86 | 75.17 |
| Expert III (SocialIQA) | 73.63 | 76.98 | 78.10 | 76.24 |
| BTM | 81.24 | 80.84 | 77.28 | 79.79 |
| ModelSoup | 80.26 | 84.25 | 77.02 | 80.51 |
| BTX | 75.76 | 81.04 | 73.39 | 76.73 |
| FlexOlmo | 75.18 | 81.11 | 76.10 | 77.46 |
| MetaMoE | **81.33** | **85.80** | **77.64** | **81.59** |

ferent scales: LLaMA-3.2-3B (Meta, 2024b), a lightweight 3B model for efficiency-critical settings, and LLaMA-3.1-8B (Meta, 2024a), a larger 8B model offering stronger performance with higher compute cost.

**Implementation Details.** For each client, we form a proxy dataset by first selecting a candidate pool of $n = 3000$ Alpaca samples most similar to its private data, then choosing $m = 500$ proxies via relevance-weighted DPP with the cosine similarity kernel $\kappa(\mathbf{x}_i, \mathbf{x}_j) = \cos(\mathbf{z}_i, \mathbf{z}_j)$, where $\mathbf{z}_i$ is $\mathbf{x}$'s embedding extracted from $\mathcal{M}_0$. Client models are initialized from the seed model $\mathcal{M}_0$. For proxy-aligned expert training, we finetune the model's FFN sublayers on both private and proxy data using LoRA (Hu et al., 2022) (rank 16, $\alpha = 32$) for 10 epochs with AdamW (Loshchilov & Hutter, 2019) (learning rate 0.0001, batch size 32, no weight decay, constant schedule with 200 warm-up steps). For router training, we adopt top-1 routing and finetune for 1 epoch with AdamW (learning rate 0.0001, batch size 32).

**Results.** Tables 3 and 4 report the testing accuracy of NLP tasks with LLaMA-3.2-3B and LLaMA-3.1-8B as the seed models, respectively. Across both backbones, MetaMoE consistently outperforms all privacy-preserving baselines. Unlike BTM, which ensembles outputs without parameter sharing, or ModelSoup, which only averages parameters, MetaMoE preserves expert specialization within a unified MoE model and enables router-based coordination. Compared with BTX and FlexOlmo, which also use proxy data but lack relevance-diverse alignment mechanisms, Meta-

MoE further benefits from DPP-selected proxies that approximate private distributions more accurately, leading to more effective routing. Quantitatively, MetaMoE achieves 74.42% average accuracy on LLaMA-3.2-3B and 81.59% on LLaMA-3.1-8B, outperforming the strongest baselines.

### 4.3. Effectiveness of Relevance-Weighted DPP

We conduct an ablation study to evaluate the effect of incorporating relevance-weighted DPP into proxy data selection (Section 3.2) for MetaMoE and the recent method FlexOlmo. The configuration denoted as "✗" in Table 5 corresponds to FlexOlmo's original similarity-based approach, which selects public samples based on their relevance scores to the client's private data, without enforcing diversity. As shown in Table 5, DPP consistently boosts accuracy across both NLP and CV tasks, highlighting its effectiveness as a selection strategy. For FlexOlmo, incorporating DPP leads to clear accuracy gains across all tasks (e.g., +0.85 on LLaMA-3.2-3B and +2.32 on LLaMA-3.1-8B), demonstrating that existing methods can benefit significantly from our proposed relevance-weighted DPP proxy selection. MetaMoE further boosts these benefits, with consistent improvements in all settings, ultimately achieving the best overall performance. These findings demonstrate that relevance-weighted DPP is essential for selecting proxy samples that effectively train the router to coordinate experts.

*Table 5.* Accuracy of FlexOlmo and MetaMoE with and without relevance-weighted (RW) DPP.

| | RW DPP | NLP | | CV | |
|---|---|---|---|---|---|
| | | LLaMA-3.2-3B | LLaMA-3.1-8B | ViT-B/32 | ViT-B/16 |
| FlexOlmo | ✗ | 72.50 | 77.46 | 92.92 | 93.53 |
| | ✓ | 73.35 | 79.78 | 93.20 | 94.38 |
| MetaMoE | ✗ | 73.60 | 80.32 | 94.12 | 95.39 |
| | ✓ | **74.42** | **81.59** | **94.52** | **96.24** |

**Visualization of Selected Proxy Samples.** In Figure 2, we use t-SNE (Van der Maaten & Hinton, 2008) to visualize the proxy samples selected by three different selection strategies: *random sampling*, *similarity-based selection* used in FlexOlmo, and our MetaMoE based on *relevance-weighted DPP*. As can be seen from Figure 2(a), random selection fails to capture either relevance or diversity, often yielding samples that are poorly aligned with the private data distribution. FlexOlmo improves alignment by selecting samples highly similar to the private domain, but the resulting proxies lack diversity: many chosen samples cluster in a narrow region of the data space, providing limited coverage of the right side of the private-data manifold (Figure 2(b)). In contrast, our MetaMoE explicitly balances relevance and diversity through the relevance-weighted DPP kernel. As shown in Figure 2(c), the resulting proxy set not only anchors closely to the private domain but also forms a diverse cover of the data space. This richer proxy distribution allows the router to discriminate across heterogeneous con-

*Table 6.* Average accuracy of MetaMoE with and without context-aware (CA) router across NLP and CV tasks.

| CA Router | NLP | | CV | |
|---|---|---|---|---|
| | LLaMA-3.2-3B | LLaMA-3.1-8B | ViT-B/32 | ViT-B/16 |
| ✗ | 72.62 | 79.41 | 93.92 | 95.94 |
| ✓ | **74.42** | **81.59** | **94.52** | **96.24** |

*Table 7.* Average accuracy of MetaMoE with and without Proxy-Aligned (PA) Expert Training across NLP and CV tasks.

| PA Training | NLP | | CV | |
|---|---|---|---|---|
| | LLaMA-3.2-3B | LLaMA-3.1-8B | ViT-B/32 | ViT-B/16 |
| ✗ | 72.71 | 80.99 | 92.98 | 94.09 |
| ✓ | **74.42** | **81.59** | **94.52** | **96.24** |

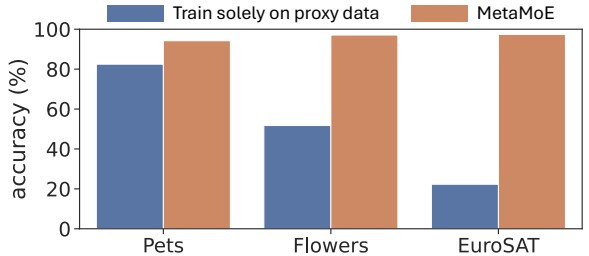

*Figure 3.* Comparison of MetaMoE with training solely on proxy data in the CV setting with CLIP ViT-B/16.

texts. Consequently, our MetaMoE provides a stronger and more aligned proxy of the unavailable private data, which translates into consistent performance gains (Table 5).

### 4.4. Ablation Study

We examine **the effectiveness of the context-aware router** (Section 3.4). As shown in Table 6, incorporating sequence-level context consistently yields higher accuracy across all evaluated datasets and backbones. These gains demonstrate that incorporating sequence-level context into the router enables more reliable expert assignment.

We assess **the effectiveness of proxy-aligned expert training** (Section 3.3) by comparing experts trained only on private data with those additionally exposed to client-specific proxy data. Experiments are conducted on both CV and NLP tasks. As shown in Table 7, incorporating proxy data yields consistent gains, suggesting proxy-aligned training enables more effective expert collaboration.

### 4.5. Effect of Expert Training without Private Data

To isolate the role of private data, we conduct an ablation on CV tasks with CLIP ViT-B/16 in which experts are trained solely on proxy samples and never observe client private data. As shown in Figure 3, training experts exclusively on proxy data results in substantial accuracy degradation across all domains. This confirms that proxy samples alone are insufficient for learning domain-specific expertise and

primarily provide coordination signals for router learning, rather than effective supervision for expert training.

In contrast, MetaMoE combines private data for expert specialization with proxy data for alignment and routing, achieving significantly higher accuracy. These results demonstrate that private-domain data are essential for learning strong experts, while proxy data play a complementary role by facilitating effective router learning.

### 4.6. Robustness to the Choice of Public Dataset

Our main NLP experiments use Alpaca (Taori et al., 2023) as the public dataset $\mathcal{D}_0$. To verify that MetaMoE does not depend on a specific public corpus, we conduct an additional experiment using OpenOrca (Lian et al., 2023) as an alternative public dataset. OpenOrca is a large-scale collection of augmented FLAN data (Longpre et al., 2023) with GPT-generated responses, differing substantially from Alpaca in both scale and construction methodology. All other settings (seed models, client datasets, hyperparameters) remain identical to those in Section 4.2.

As shown in Table 8, MetaMoE consistently outperforms all baselines when using OpenOrca, confirming that a general-purpose public dataset suffices without careful domain-specific curation. Combined with the results using Alpaca and the ImageNet experiment on CV tasks (Section 4.1), these results demonstrate that MetaMoE is robust across diverse public dataset choices in both modalities.

*Table 8.* Accuracy of NLP tasks using OpenOrca as the public dataset with LLaMA-3.2-3B as the seed model.

| | CSQA | CosmosQA | SocialIQA | **Average** |
|---|---|---|---|---|
| ModelSoup | 73.71 | 75.24 | 71.75 | 73.57 |
| BTM | 74.61 | 75.44 | 68.17 | 72.74 |
| BTX | 71.66 | 72.90 | 69.40 | 71.32 |
| FlexOlmo | 73.63 | 73.97 | 70.73 | 72.78 |
| MetaMoE | **75.18** | **77.12** | **72.06** | **74.79** |

### 5. Conclusion

We proposed MetaMoE, a privacy-preserving framework for unifying independently trained experts into a single Mixture-of-Experts model. By leveraging diversity-aware proxy selection for router learning and proxy-aligned expert training, MetaMoE enables effective expert coordination without sharing private data. We further provide formal privacy guarantees showing that the shared routing vectors have bounded per-sample sensitivity and do not reveal recoverable domain-level information about private data. Experiments on computer vision and natural language processing benchmarks show consistent improvements over recent methods, underscoring the importance of diversity-aware proxies for privacy-preserving MoE unification.

## Acknowledgements

The research work described in this paper was conducted in the JC STEM Lab of Machine Learning and Symbolic Reasoning funded by The Hong Kong Jockey Club Charities Trust.

## Impact Statement

This paper proposes a framework for privacy-preserving model unification. While MetaMoE enforces data residency and provides formal guarantees on the shared routing vectors, deploying it in high-stakes domains (e.g., healthcare, finance) may require additional safeguards such as formal differential privacy mechanisms or external audits. We encourage practitioners to evaluate the privacy properties of all communicated artifacts in the context of their specific regulatory requirements.

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

## A. Computation of Relevance Score

Following FlexOlmo (Shi et al., 2025), we compute the relevance score $g(\mathbf{x}, \mathcal{D}_p)$ of a public sample $\mathbf{x} \in \mathcal{D}_0$ with respect to a client dataset $\mathcal{D}_p$ by training a binary classifier to distinguish $\mathcal{D}_p$ from $\mathcal{D}_0$. Specifically, we construct a training set by labeling samples from $\mathcal{D}_p$ as positive and randomly drawing 10K samples from $\mathcal{D}_0$ as negative. We append a classification head to the last hidden layer of the seed model, and finetune this classifier to distinguish whether a sample comes from the client dataset $\mathcal{D}_p$ (positive) or the public dataset $\mathcal{D}_0$ (negative). After training, we apply the classifier to every $\mathbf{x} \in \mathcal{D}_0$. The predicted probability that $\mathbf{x}$ belongs to $\mathcal{D}_p$ is used as the relevance score:

$$g(\mathbf{x}, \mathcal{D}_p) = \mathbb{P}[\text{classifier predicts } \mathbf{x} \in \mathcal{D}_p]. \tag{10}$$

This score quantifies the extent to which each public sample is representative of the private domain. In practice, we rank all $\mathbf{x} \in \mathcal{D}_0$ by $g(\mathbf{x}, \mathcal{D}_p)$ and retain the top-$n$ candidates for subsequent proxy selection.

## B. Cholesky Updates for Efficient Inference

In this appendix, we show how to make greedy DPP MAP inference computationally efficient for large-scale proxy selection. The main challenge is the repeated evaluation of determinants for growing kernel submatrices. A naive implementation recomputes a full factorization for each of the $n$ candidates at every greedy step, incurring $\mathcal{O}(nm^3)$ time per iteration when the current subset has size $m$. We address this bottleneck by maintaining a Cholesky factorization of the current kernel submatrix and updating it incrementally as new elements are considered. With this strategy, scoring all $n$ candidates in a greedy iteration requires only $\mathcal{O}(nm)$ time. The derivation below details this update mechanism and explains its implications for scalability.

**Naive Computation.** Consider a candidate subset $\mathcal{S} \subseteq \{1, \ldots, n\}$ with associated kernel submatrix $\widetilde{\mathbf{L}}_{\mathcal{S}}$. At each step of greedy MAP inference, one must compute

$$\det(\widetilde{\mathbf{L}}_{\mathcal{S} \cup \{\mathbf{x}\}})$$

for a new candidate $\mathbf{x} \notin \mathcal{S}$. If performed directly, this requires recomputing the determinant of an $(|\mathcal{S}| + 1) \times (|\mathcal{S}| + 1)$ matrix, incurring $\mathcal{O}(m^3)$ time per evaluation. Over all $n$ candidates and $m$ greedy steps, the total cost for each greedy step scales as $\mathcal{O}(nm^3)$, which is infeasible when both $n$ and $m$ are large.

**Cholesky Factorization.** To avoid redundant recomputation, we exploit the Cholesky decomposition (Horn & Johnson, 1985). Suppose the current kernel submatrix admits a decomposition

$$\widetilde{\mathbf{L}}_{\mathcal{S}} = \mathbf{P}\mathbf{P}^{\top},$$

where $\mathbf{P}$ is a lower-triangular matrix of size $|\mathcal{S}| \times |\mathcal{S}|$. The determinant is then easily obtained as

$$\det(\widetilde{\mathbf{L}}_{\mathcal{S}}) = \prod_{i=1}^{|\mathcal{S}|} \mathbf{P}_{ii}^2,$$

so that the computational burden shifts from determinant computation to maintaining $\mathbf{P}$.

**Incremental Update.** When a new element $\mathbf{x}$ is considered, the augmented kernel matrix can be written in block form:

$$\widetilde{\mathbf{L}}_{\mathcal{S} \cup \{\mathbf{x}\}} = \begin{bmatrix} \widetilde{\mathbf{L}}_{\mathcal{S}} & \mathbf{k} \\ \mathbf{k}^{\top} & \widetilde{\mathbf{L}}_{\mathbf{xx}} \end{bmatrix}, \tag{11}$$

where $\mathbf{k}$ contains the similarities between $\mathbf{x}$ and items in $\mathcal{S}$. Instead of recomputing a full factorization, we extend $\mathbf{P}$ by one row and column:

$$\mathbf{P}' = \begin{bmatrix} \mathbf{P} & 0 \\ \mathbf{y}^{\top} & \sigma \end{bmatrix},$$

so that $\mathbf{P}'\mathbf{P}'^{\top} = \widetilde{\mathbf{L}}_{\mathcal{S} \cup \{\mathbf{x}\}}$. Expanding both sides and equating with (11) gives

$$\mathbf{P}\mathbf{y} = \mathbf{k}, \qquad \sigma^2 = \widetilde{\mathbf{L}}_{\mathbf{xx}} - \|\mathbf{y}\|_2^2.$$

Thus, the update reduces to solving a triangular system (for $\mathbf{y}$) and computing a residual variance (for $\sigma^2$). Both steps are efficient: solving a triangular system costs $\mathcal{O}(m)$, and computing a norm is linear in $m$ as well. Hence, each iteration of greedy MAP inference only requires a total time of $\mathcal{O}(nm)$ for searching over the $n$ candidates. This makes the approach scalable to large public datasets, while still retaining the DPP's balance between relevance and diversity.

## C. Computer Vision Datasets

Figure 4 presents randomly sampled images from the three client datasets used in the CV experiments: Pets (Parkhi et al., 2012), Flowers (Nilsback & Zisserman, 2008), and EuroSAT (Helber et al., 2019). These examples illustrate the visual diversity across domains, ranging from fine-grained object recognition of dog and cat breeds (Pets), to natural scene categorization of flower species (Flowers), and remote sensing imagery for land-use classification (EuroSAT). Such heterogeneity highlights the challenge of unifying domain-specialized experts into a single model while preserving privacy and ensuring robust multi-domain generalization.

We adopt ImageNet (Deng et al., 2009) as the public dataset from which proxy samples are drawn. Figure 5 shows randomly sampled examples from ImageNet.

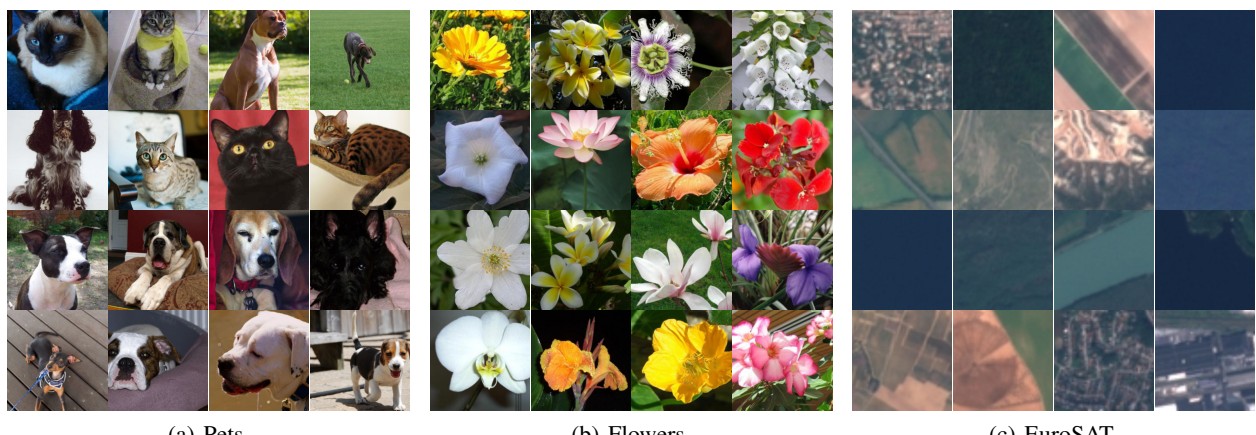

(a) Pets.        (b) Flowers.        (c) EuroSAT.

*Figure 4.* Sample images from the three client domains: Pets, Flowers, and EuroSAT.

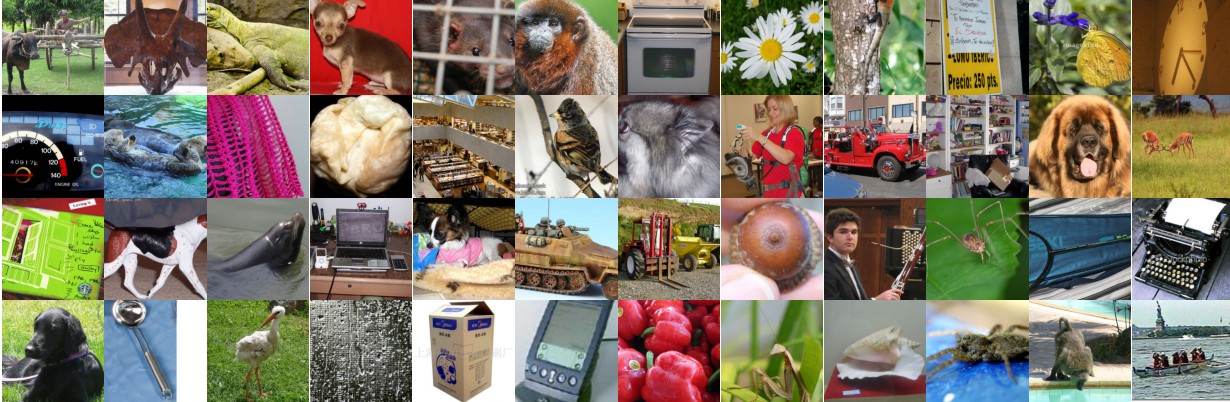

*Figure 5.* Sample images from ImageNet.

## D. Natural Language Processing Datasets

The client-side NLP datasets comprise CommonsenseQA (Talmor et al., 2019), CosmosQA (Huang et al., 2019), and SocialIQA (Sap et al., 2019). These cover complementary reasoning skills: CommonsenseQA requires grounding abstract questions in everyday knowledge; CosmosQA emphasizes multi-sentence comprehension with causal and temporal reasoning; and SocialIQA targets social motivations and reactions in human interactions. Examples 1–3 show two representative

samples from CommonsenseQA, CosmosQA, and SocialIQA, respectively. Such diversity highlights the challenge of integrating domain-specialized experts in NLP while ensuring broad generalization across reasoning styles and task formats.

As the public corpus $\mathcal{D}_0$, we use Alpaca (Taori et al., 2023), an open-domain instruction–response dataset ranging from factual queries to reasoning and generation. Example 4 illustrates three instances from Alpaca.

---

**Example 1: Samples from CommonsenseQA (Talmor et al., 2019)**

**Question:** The fox walked from the city into the forest, what was it looking for?
(A) pretty flowers
(B) hen house
(C) natural habitat
(D) storybook
(E) dense forest
**Answer:** (C) natural habitat

**Question:** To learn one must have the right book, to work efficiently what must one have?
(A) improve yourself
(B) become knowledgeable
(C) have tools
(D) persistence
(E) have more knowledge
**Answer:** (C) have tools

---

**Example 2: Samples from CosmosQA (Huang et al., 2019)**

Let me be clear, everything is good. Having said that, I 've been a little preoccupied lately, as Leslie threw out her back on Sunday. It's been an interesting, and marriage-solidifying 36 hours. Sunday afternoon, around 2 pm, Leslie and I were hanging in the library, sans kids (they were with their father), listening to music and talking.
**Question:** How did Leslie's throwing out of her back help to solidify your marriage?
(A) Leslie realized that I am really good at taking care of people and wants to keep me around for that reason
(B) Leslie saw that if she ever hurt herself she would need my help to heal and realized that she needs me
(C) None of the above choices
(D) We were able to spend time together just talking and that helped us to connect better
**Answer:** (D) We were able to spend time together just talking and that helped us to connect better

But the thing is, I forgave and forgot a while ago. I 'm not bitter anymore. I just genuinely don't want to go back to that church. I couldn't get the balls to tell her that ...
**Question:** Why wouldn't I have the balls to tell her that?
(A) I am not a man and I can't tell her those things to her face
(B) I am afraid of that church and I would feel less manly if I told her
(C) None of the above choices
(D) It is her favorite church and I don't want to hurt her feelings
**Answer:** (D) It is her favorite church and I don't want to hurt her feelings

---

**Example 3: Samples from SocialIQA (Sap et al., 2019)**

During their yearbook class, Alex took an old photo of herself and Quinn, and crossed out Quinn's face on the photograph. Quinn saw her do this.
**Question:** How would Quinn feel as a result?
(A) Like Alex misses hanging out with her
(B) Like Alex betrayed her
(C) be friends again
**Answer:** (B) Like Alex betrayed her

Tracy was teaching Kendall to dance so Tracy raised Kendall's hips into the right position for the next move.
**Question:** How would you describe Tracy?
(A) a bad teacher
(B) hopeful that this doesn't make Kendall uncomfortable
(C) a patient teacher
**Answer:** (B) hopeful that this doesn't make Kendall uncomfortable

> **Example 4: Samples from Alpaca (Taori et al., 2023)**
>
> **Question:** Use the given data to calculate the median.
> **Input:** [2, 3, 7, 8, 10]
> **Answer:** The median is 7.
>
> **Question:** Analyze the given text for its tone.
> **Input:** The world has been greatly impacted by the COVID-19 pandemic and it has drastically changed our lives.
> **Answer:** The tone of the text is one of concern and recognition of the dramatic effects of the COVID-19 pandemic on the world. It is an acknowledgement of the upheaval that the virus has caused and that our lives have been drastically changed.
>
> **Question:** Explain the concept of artificial intelligence in simple terms.
> **Answer:** Artificial Intelligence (AI) is the ability of machines and computers to make decisions and complete tasks similar to what humans can do. AI can learn from data, recognize patterns, and make predictions. AI technology can automate tedious and complex processes, helping to reduce human effort and errors, and enable more accurate and efficient decisions. AI can be used in a wide range of applications, from robotics and autonomous vehicles to healthcare. It is increasingly becoming an integral part of everyday life.

## E. Additional Experiments

### E.1. Computation Overhead of Relevance-Weighted DPP

Both FlexOlmo and MetaMoE require computing embeddings for all public samples, which dominates the proxy-selection cost. The additional work unique to MetaMoE is the greedy MAP inference for relevance-weighted DPP, implemented with efficient Cholesky updates (see Appendix B). Table 9 shows the measured running time for NLP and CV setups. Relevance-weighted DPP adds only 0.2–0.3 minutes (3–5%) per client, while expert finetuning takes over 10 GPU-hours, confirming that the diversity-aware selection introduces only marginal overhead relative to similarity-only baselines while delivering more representative proxy sets.

*Table 9.* Proxy-selection running time (minutes).

|  | NLP | | CV | |
| --- | --- | --- | --- | --- |
|  | LLaMA-3.2-3B | LLaMA-3.1-8B | ViT-B/32 | ViT-B/16 |
| Similarity-based selection | 1.91 | 4.21 | 6.41 | 15.70 |
| Relevance-weighted DPP | 2.17 | 4.54 | 6.61 | 15.91 |

### E.2. Comparison with Federated Learning Methods: CoMiGS (Fan et al., 2025) and MoA (Feng et al., 2024)

Federated learning (FL) methods differ fundamentally from MetaMoE: they require exchanging large model states every round. This leads to substantial bandwidth and memory overhead and creates instability when client data are heterogeneous because divergent local updates must be averaged. MetaMoE eliminates synchronization entirely—each client fine-tunes its expert locally (on private plus proxy data), and only a single exchange of frozen expert weights occurs before router training.

**Comparison with CoMiGS (Fan et al., 2025).** CoMiGS (Fan et al., 2025) adopts a federated learning paradigm that requires repeated synchronization among clients—periodically exchanging model parameters for joint optimization. This approach incurs high communication and memory costs and often becomes unstable under heterogeneous client data, leading to degraded performance. In contrast, MetaMoE avoid exchanging model parameters entirely. Each client independently fine-tunes its expert on private and proxy data, and all experts are unified once through MoE integration. This merge-after-training design achieves better scalability, eliminates communication overhead, and remains stable under heterogeneous data.

**Comparison with Mixture-of-LoRAs (MoA) (Feng et al., 2024).** MoA also seeks to unify multiple LoRA experts but assumes direct access to all client data for router training, which violates privacy constraints. In the experiments reported here, MoA is reimplemented by training its router only on public data to maintain expert coordination without violating privacy constraints, enabling a fair comparison under the same privacy-preserving setting.

**Empirical results.** MetaMoE is compared with CoMiGS and MoA under the same LoRA configurations. As shown in Table 10, MetaMoE consistently outperforms both methods across all benchmarks, demonstrating that MetaMoE's unification offers better performance.

*Table 10.* Comparison with CoMiGS and MoA on NLP tasks (LLaMA-3.2-3B).

|  | CSQA | CosmosQA | SocialIQA | Average |
|---|---|---|---|---|
| CoMiGS (Fan et al., 2025) | 72.32 | 71.46 | 71.19 | 71.65 |
| MoA (Feng et al., 2024) | 71.09 | 74.64 | 70.11 | 71.95 |
| MetaMoE | **74.94** | **76.05** | **72.26** | **74.42** |

### E.3. Effect of Expert Training without Private Data

To isolate the role of private data, we conduct an ablation where experts are trained solely on proxy samples and never observe client data. We evaluate this proxy-only baseline in the CV setting. As summarized in Tables 11 and 12, training exclusively on proxies leads to large accuracy drops, confirming that proxies primarily provide coordination signals for router learning. MetaMoE, which blends private data (for domain expertise) with proxies (for alignment), substantially outperforms the proxy-only alternative, demonstrating the necessity of private-domain expertise.

*Table 11.* Proxy-only baseline versus MetaMoE (CLIP ViT-B/32).

|  | Pets | Flowers | EuroSAT | Average |
|---|---|---|---|---|
| Train solely on proxy data | 78.60 | 42.63 | 12.98 | 44.74 |
| MetaMoE | **91.91** | **93.67** | **97.98** | **94.52** |

*Table 12.* Proxy-only baseline versus MetaMoE (CLIP ViT-B/16).

|  | Pets | Flowers | EuroSAT | Average |
|---|---|---|---|---|
| Train solely on proxy data | 82.47 | 51.73 | 22.30 | 52.17 |
| MetaMoE | **94.22** | **97.08** | **97.41** | **96.24** |

### E.4. Privacy-Preserving Guarantees

MetaMoE never exposes private data, and any similarity between selected proxy samples and private data does not constitute a privacy violation, as all proxy candidates are drawn from a public dataset that is already accessible to all parties.

**(1) Semantic similarity is not equivalent to private-data exposure.** The relevance-weighted DPP method selects public samples that are representation-wise similar to the client domain. These proxies may resemble private data but are not derived from private samples, and all clients already have access to them. In privacy-preserving systems, leakage occurs only when *non-public information becomes newly revealed*. Selecting an already public sample—no matter how similar—does not expose any new private information.

**(2) Overlap with public data does not constitute private-data exposure.** If a public sample coincidentally overlaps with one in a client's dataset, revealing that sample still constitutes *public-data exposure*, not private-data exposure, since the content is already publicly available prior to any interaction with MetaMoE. This principle aligns with standard privacy frameworks such as the California Consumer Privacy Act (CCPA), which explicitly excludes "publicly available information" from the definition of personal data. Under this definition, *the exposure of a public sample is not regarded as a violation of privacy*.

**(3) MetaMoE introduces no new channels for private-data leakage.** The proxy-selection process transmits only the IDs of selected public samples and the final expert weights—never private samples, gradients, or intermediate activations. Because all proxy candidates are drawn from a public dataset, the process does not disclose or allow inference about private data.

### E.5. Comparison with LoRASuite (Li et al., 2025)

MetaMoE is a low-rank adaptation unification method, whereas LoRASuite (Li et al., 2025) is not, and the two address fundamentally different goals. MetaMoE aims to *unify multiple domain-specialized LoRA experts* trained on the same backbone into a privacy-preserving MoE—answering how to combine many LoRA experts without sharing client data. In contrast, LoRASuite focuses on *LoRA migration*, transferring a single LoRA adapter trained on backbone (A) to backbone

(B) after the backbone is upgraded, with the goal of expert adaptation rather than expert unification. Furthermore, LoRASuite requires *client data* to align activations and performs poorly without it, while MetaMoE assumes *no private data access* and relies entirely on public proxies for supervision. Given these fundamental differences in objectives and data requirements, a direct empirical comparison would be inappropriate, as it would force LoRASuite into a multi-expert unification setting it was never designed for.

### E.6. Computational Cost Analysis

This section provides a detailed comparison of MetaMoE's computational cost against existing methods, covering both unification time and inference speed across vision and language tasks. The results demonstrate that MetaMoE achieves strong accuracy improvements without adding significant overhead.

**Unification cost.** The merging (unification) time of MetaMoE is nearly identical to FlexOlmo and BTX across all backbones, showing that the accuracy gains do not come from higher computational cost during unification. While BTM and ModelSoup appear faster (or cost-free), they avoid the coordination required for MoE merging, which explains their lower accuracy. The small extra cost in MetaMoE is therefore a modest and worthwhile trade-off for its significant accuracy improvement.

**Inference efficiency.** MetaMoE maintains inference speeds comparable to other MoE unification methods, confirming that the context-aware router introduces minimal runtime overhead and scales efficiently across backbones. In contrast, BTM requires inference over all experts for every input, leading to approximately $3\times$ slower inference despite its zero unification cost.

**Overall cost–performance balance.** Across both CV and NLP tasks, MetaMoE consistently achieves state-of-the-art accuracy with comparable computational efficiency, as summarized in Tables 13 and 14.

*Table 13.* Cost–performance comparison on CV tasks.

| | ViT-B/32 | | | ViT-B/16 | | |
|---|---|---|---|---|---|---|
| | ACC | Unify Time (s) | Inference Speed (samples/s) | ACC | Unify Time (s) | Inference Speed (samples/s) |
| BTM | 90.33 | – | 606 | 91.75 | – | 249 |
| ModelSoup | 74.20 | 5.72 | 1813 | 79.42 | 5.72 | 743 |
| BTX | 74.30 | 11.13 | 1758 | 81.20 | 19.72 | 715 |
| FlexOlmo | 92.92 | 11.93 | 1767 | 93.53 | 18.24 | 719 |
| MetaMoE | **94.52** | 12.15 | 1751 | **96.24** | 19.88 | 710 |

*Table 14.* Cost–performance comparison on NLP tasks.

| | LLaMA-3.2-3B | | | LLaMA-3.1-8B | | |
|---|---|---|---|---|---|---|
| | ACC | Unify Time (s) | Inference Speed (samples/s) | ACC | Unify Time (s) | Inference Speed (samples/s) |
| BTM | 72.74 | – | 17.44 | 79.79 | – | 8.20 |
| ModelSoup | 73.57 | 8.24 | 43.46 | 80.51 | 11.25 | 22.86 |
| BTX | 71.14 | 118.21 | 42.12 | 76.73 | 223.37 | 21.41 |
| FlexOlmo | 72.50 | 119.90 | 41.59 | 77.46 | 206.28 | 21.95 |
| MetaMoE | **74.42** | 114.46 | 40.67 | **81.59** | 205.42 | 20.05 |

### E.7. Robustness under Non-Overlapping Public Data

In our main CV experiments (Section 4.1), the public dataset ImageNet may share semantic overlap with certain client domains (e.g., dog/cat breeds for Pets, flower species for Flowers). To evaluate MetaMoE under a more challenging setting where such overlap is eliminated, we remove all ImageNet categories that are semantically related to the three client domains, including dog and cat breeds (for Pets), flower species (for Flowers), and satellite-like or aerial imagery (for EuroSAT). All other settings remain identical to those in Section 4.1 with CLIP ViT-B/32.

As shown in Table 15, MetaMoE remains the strongest method under zero domain overlap, achieving $93.78\%$ average accuracy with only a $0.74$-point drop compared to the full-overlap setting ($94.52\%$, Table 1). In contrast, FlexOlmo degrades by $3.53$ points ($89.39\%$ vs. $92.92\%$), confirming that relevance-weighted DPP is more robust to domain gap than relevance-only proxy selection. Moreover, the NLP experiments in the main paper (Section 4.2) also reflect a non-overlapping regime:

the public dataset Alpaca shares no domain overlap with the client datasets by construction, yet MetaMoE consistently outperforms all baselines (Tables 3–4).

*Table 15.* Accuracy on CV tasks with CLIP ViT-B/32 under non-overlapping public data, where all ImageNet categories semantically related to client domains are removed.

|  | Pets | Flowers | EuroSAT | **Average** |
|---|---|---|---|---|
| ModelSoup | 87.90 | 70.52 | 64.19 | 74.20 |
| BTM | 90.81 | 85.10 | 95.07 | 90.33 |
| BTX | 80.35 | 61.10 | 58.16 | 66.54 |
| FlexOlmo | 86.56 | 88.55 | 93.07 | 89.39 |
| MetaMoE | **91.01** | **92.53** | **97.80** | **93.78** |

# F. Privacy Analysis of Routing Vectors

This section provides the complete derivations for the privacy guarantees stated in Section 3.6. We analyze the routing vector $\mathbf{e}_p^{(l)}$ shared by each client $p$ and show that it reveals negligible private information, with formal guarantees that are independent of both the domain gap and the private dataset size.

**Notation.** We fix a client $p$ and a layer $l$, and drop the subscript $p$ and superscript $(l)$ when context is clear. Let $\{\mathbf{x}_i\}_{i=1}^N$ denote the private samples in $\mathcal{D}_p$ and $\{\mathbf{z}_j\}_{j=1}^m$ denote the proxy samples in $\hat{\mathcal{D}}_p$, where $N = |\mathcal{D}_p|$ and $m = |\hat{\mathcal{D}}_p|$. Let $f(\cdot) = \mathcal{M}_p^{(1:l)}(\cdot)$ denote the encoder (the first $l$ layers of expert $\mathcal{M}_p$), and let $B = \max_{\mathbf{x}} \|f(\mathbf{x})\|_2$ denote the embedding norm bound. Define the mean private embedding and the mean proxy embedding as

$$\boldsymbol{\mu}_{\text{priv}} = \frac{1}{N} \sum_{i=1}^N f(\mathbf{x}_i), \qquad \boldsymbol{\mu}_{\text{proxy}} = \frac{1}{m} \sum_{j=1}^m f(\mathbf{z}_j). \tag{12}$$

The routing vector $\mathbf{e}_p^{(l)}$ (defined in (7)) can then be written as

$$\mathbf{e} = \frac{1}{N+m} \left( \sum_{i=1}^N f(\mathbf{x}_i) + \sum_{j=1}^m f(\mathbf{z}_j) \right) = \frac{N}{N+m} \boldsymbol{\mu}_{\text{priv}} + \frac{m}{N+m} \boldsymbol{\mu}_{\text{proxy}}. \tag{13}$$

**Adversary's knowledge.** We consider an honest-but-curious adversary (e.g., the central server) that can observe the routing vector $\mathbf{e}$ and knows the public quantities $\boldsymbol{\mu}_{\text{proxy}}$, $m$, and $f$. Crucially, the adversary **does not know** $N$, the private dataset size, as it is never communicated by the MetaMoE protocol.

## F.1. Per-Sample Sensitivity Bound

We show that the influence of any single private sample on the routing vector is bounded by $O(1/m)$, regardless of $N$ and the domain gap.

**Proposition F.1** (Sensitivity bound). *Let $\mathbf{e}$ be the routing vector defined in (13), and let $\mathbf{e}'$ denote the routing vector obtained by replacing a single private sample $\mathbf{x}_k$ with an arbitrary sample $\mathbf{x}_k'$. The $\ell_2$ sensitivity of $\mathbf{e}$ satisfies*

$$\Delta_2(\mathbf{e}) = \max_{\mathbf{x}_k, \mathbf{x}_k'} \|\mathbf{e} - \mathbf{e}'\|_2 \leq \frac{2B}{N+m} \leq \frac{2B}{m}. \tag{14}$$

*Proof.* Replacing $\mathbf{x}_k$ with $\mathbf{x}_k'$ changes only the $k$-th term in the private sum, yielding

$$\mathbf{e} - \mathbf{e}' = \frac{1}{N+m} \big( f(\mathbf{x}_k) - f(\mathbf{x}_k') \big). \tag{15}$$

Taking the $\ell_2$ norm and applying the triangle inequality gives

$$\|\mathbf{e} - \mathbf{e}'\|_2 = \frac{\|f(\mathbf{x}_k) - f(\mathbf{x}_k')\|_2}{N+m} \leq \frac{\|f(\mathbf{x}_k)\|_2 + \|f(\mathbf{x}_k')\|_2}{N+m} \leq \frac{2B}{N+m}. \tag{16}$$

Since $N \geq 0$ implies $N + m \geq m$, we obtain $\Delta_2(\mathbf{e}) \leq 2B/m$. $\qquad \square$

**Implications.** The bound $\Delta_2(\mathbf{e}) \le 2B/m$ depends solely on the proxy set size $m$, which is a public hyperparameter. In our experiments, $m = 500$, so each individual private sample's contribution is diluted among at least 500 embeddings, making its influence on $\mathbf{e}$ negligibly small. Furthermore, this bound is independent of both the private dataset size $N$ and the domain gap between private and proxy data. When $N$ is small (a challenging privacy scenario), the bound is already controlled by $O(1/m)$; when $N$ grows large, it additionally tightens to $O(1/(N + m))$.

This sensitivity bound is stated in the same sense as the foundational framework of differential privacy (Dwork & Roth, 2014): low sensitivity implies that any individual private sample has a vanishingly small effect on the shared statistic, which is a necessary condition for strong privacy guarantees.

### F.2. Unrecoverability of Private-Data Statistics

We show that the mean private embedding $\boldsymbol{\mu}_{\text{priv}}$, the coarsest possible summary of the private data, cannot be recovered from the routing vector $\mathbf{e}$.

**Proposition F.2** (Unrecoverability). *Given access to the routing vector $\mathbf{e}$, the mean proxy embedding $\boldsymbol{\mu}_{\text{proxy}}$, the proxy set size $m$, and the encoder $f$, an adversary cannot uniquely determine $\boldsymbol{\mu}_{\text{priv}}$ without knowledge of the private dataset size $N$.*

*Proof.* From (13), the adversary can compute

$$(N + m)\,\mathbf{e} - m\,\boldsymbol{\mu}_{\text{proxy}} = N\,\boldsymbol{\mu}_{\text{priv}}. \tag{17}$$

Isolating $\boldsymbol{\mu}_{\text{priv}}$ yields

$$\boldsymbol{\mu}_{\text{priv}} = \frac{(N + m)\,\mathbf{e} - m\,\boldsymbol{\mu}_{\text{proxy}}}{N}. \tag{18}$$

Since the adversary knows $\mathbf{e}$, $\boldsymbol{\mu}_{\text{proxy}}$, and $m$, evaluating (18) requires knowing $N$, which appears both in the numerator and the denominator and cannot be canceled out. Because $N$ is never communicated by the MetaMoE protocol, the adversary cannot evaluate (18) for the true value of $N$, and therefore cannot recover $\boldsymbol{\mu}_{\text{priv}}$. □

**Implications.** The mean private embedding $\boldsymbol{\mu}_{\text{priv}}$ represents the coarsest possible form of domain-level information: it is a single vector that summarizes the entire private dataset's representation in the encoder's feature space. Since even this coarsest statistic is unrecoverable from $\mathbf{e}$, finer-grained distributional properties of $\mathcal{D}_p$ (e.g., variance, class proportions, cluster structure, or individual samples) are a fortiori unrecoverable. This directly resolves the concern on whether the routing vector could leak domain-level or distributional information about the private data.

### F.3. Comparison with FlexOlmo

We show that MetaMoE exposes strictly less private information than FlexOlmo (Shi et al., 2025) through the routing mechanism.

FlexOlmo initializes its domain-informed router using per-expert routing embeddings computed as the mean embedding over *private data alone* (Section 3.3.2 of Shi et al. (2025)). Concretely, FlexOlmo shares the vector $\boldsymbol{\mu}_{\text{priv}} = \frac{1}{N} \sum_{i=1}^{N} f(\mathbf{x}_i)$, which is the complete mean private embedding. In contrast, MetaMoE shares $\mathbf{e} = \frac{N}{N+m}\,\boldsymbol{\mu}_{\text{priv}} + \frac{m}{N+m}\,\boldsymbol{\mu}_{\text{proxy}}$ (13), where the private component $\boldsymbol{\mu}_{\text{priv}}$ is diluted by averaging with the publicly computable $\boldsymbol{\mu}_{\text{proxy}}$.

This comparison establishes the following:

(i) **FlexOlmo directly reveals $\boldsymbol{\mu}_{\text{priv}}$**, giving an adversary full access to the mean private embedding.

(ii) **MetaMoE never reveals $\boldsymbol{\mu}_{\text{priv}}$ in isolation.** The adversary observes only the diluted mixture $\mathbf{e}$, from which $\boldsymbol{\mu}_{\text{priv}}$ is unrecoverable without $N$ (Proposition F.2).

(iii) **Per-sample sensitivity is also stronger.** FlexOlmo's routing embedding has sensitivity $2B/N$, which grows large when $N$ is small. MetaMoE's sensitivity is $2B/(N + m) \le 2B/m$, which remains bounded even for small $N$.

Therefore, MetaMoE provides a strictly stronger privacy guarantee than FlexOlmo in terms of both the recoverability of private-data statistics and the per-sample sensitivity.

