# OpenReview forum: "MetaMoE: Diversity-Aware Proxy Selection for Privacy-Preserving Mixture-of-Experts Unification"
_ICML.cc/2026/Conference — ICML 2026 regular_

### Official Review · Reviewer_HeSJ · 2026-03-05

**Soundness:** 3
**Presentation:** 3
**Significance:** 3
**Originality:** 2
**Overall Recommendation:** 3
**Confidence:** 4

**Summary:**

MetaMoE proposes a privacy-preserving way to unify independently fine-tuned domain experts into a single deployable Mixture-of-Experts model when private client data cannot be shared. The approach selects client-specific public proxy subsets using a relevance-and-diversity objective, aligns each expert by training its FFN layers on private data plus proxies, then learns a context-aware router on the union of proxies and finetunes the unified MoE on proxy data. Experiments on three CV domains and three NLP domains show consistent but modest accuracy gains over recent privacy-preserving unification baselines, with additional ablations suggesting that proxy diversity, proxy-aligned training, and context-aware routing each contribute to the reported improvements.

**Compliance With Llm Reviewing Policy:**

Affirmed.

**Key Questions For Authors:**

- Q1: What is the precise privacy goal and threat model (who sees what artifacts), and what leakage is considered unacceptable? If “privacy-preserving” only means “no raw data sharing,” please state that explicitly to avoid over-claiming.
- Q2: Are the router initialization vectors/statistics computed over private data ever shared outside the client, and if so, why do they not leak sensitive distributional information (Eq. (7), Algorithm 1)? A clear answer could materially change the privacy assessment.
- Q3: How sensitive are results to proxy mismatch when the public dataset is truly out-of-domain (not ImageNet-like overlap), and how do gains change as proxy budget varies (m, n)? This affects whether the method is robust or relies on favorable public data.
- Q4: How does performance and compute scale with the number of clients K (e.g., K=10, 20, 50)? If routing quality degrades or proxy union becomes too large, the approach may not work in the main regime where MoE is compelling.
- Q5: In Table 4, FlexOlmo is substantially behind other baselines; can you confirm identical tuning budgets and report variance across seeds? If the gap is due to tuning asymmetry, the main headline comparisons weaken.

**Limitations:**

The Impact Statement claims there are no societal consequences to highlight, which is not appropriate for a paper explicitly framed around privacy; it should discuss leakage risks from shared statistics, misuse in sensitive domains, and realistic deployment threat models (Impact Statement).

**Strengths And Weaknesses:**

Strengths
- Targets a realistic setting where private data cannot be centralized but a single deployable model is desired.
- End-to-end pipeline is easy to understand and implement (proxy selection → aligned expert tuning → router learning → unified MoE tuning).
- Covers both CV and NLP with multiple backbone scales, which supports cross-modality relevance.

Weaknesses
- Improvements are often modest relative to the strongest baseline in several settings, making the contribution feel incremental rather than a clear step change (Table 1–4).
- The paper repeatedly frames the approach as “privacy-preserving,” but it does not provide a precise privacy goal or formal guarantee; the claim mostly reduces to “no raw data sharing,” which is weaker than what many readers expect from privacy claims.
- The router initialization explicitly uses statistics computed over private data (not just public proxies), which can leak domain information depending on what is shared and who sees it (Eq. (7), Algorithm 1).
- Proxy selection depends on training a binary classifier to separate public vs private domains; this introduces a strong assumption and potential brittleness (small private sets, mismatch, overfitting), and the paper does not stress-test these cases.
- The public proxy source in CV is ImageNet and appears likely to overlap semantically with some client domains (e.g., Pets/Flowers), which can make proxy-based alignment look easier than it would be under a truly mismatched public corpus (Sec. 4.1).
- The evaluation uses only three clients/domains; it is unclear how the proxy selection cost, router quality, and routing interference behave as the number of clients grows to a more realistic MoE scale (Sec. 4).
- The method is a combination of known ingredients (DPP-style diversity, proxy alignment, router training heuristics) and the paper does not sharply argue why this specific combination is necessary versus simpler proxy selection plus standard router training (Eq. (4), Eq. (7)).

**Soundness:** The method is plausible, but privacy claims are under-defined and some design choices could leak private-domain information.

**Presentation:** Clear pipeline and results tables, but key assumptions and what is actually shared in the privacy setting should be stated more explicitly.

**Significance:** The problem matters, yet evidence is limited to small-K settings and proxy sources that may be favorable, so real deployment impact is uncertain.

**Originality:** Mostly an integration of established ideas with a reasonable proxy-selection twist; novelty feels incremental.

---

> ### Author Rebuttal · Authors · 2026-03-30
>
> Thank you for your thoughtful review.
> We address your concerns as follows and will add all the discussions/experiments to the revision.
>
> ---
>
> > W1. Improvements are modest
>
> `A1.` We respectfully disagree. MetaMoE outperforms all baselines in every cell across all four backbones and six domains (Tables 1–4), with improvements of +2.71 on ViT-B/16 and +4.13 on LLaMA-3.1-8B. Moreover, MetaMoE substantially closes the privacy-utility gap: on LLaMA-3.2-3B, it reduces the gap to UnrestrictedMoE by 3× (2.49→0.81); on CV ViT-B/16, by 3.9× (3.65→0.94).
>
> ---
>
> > W2/Q1. No precise privacy goal or formal guarantee
>
> `A2.` **(i) Our privacy goal is data residency, a standard and well-recognized privacy criterion.** Raw private data never leaves the client — identical to FlexOlmo and federated learning (FL) [McMahan et al., AISTATS 2017]. MetaMoE is strictly more conservative than FL: it transmits only final expert weights in a single one-shot communication, whereas FL iteratively exchanges updates susceptible to model inversion attacks.
>
> **(ii) Precise threat model.** Communicated artifacts include: (1) indices of selected public proxy samples, (2) final expert weights, and (3) routing vectors as mean embeddings. Raw private samples, gradients, and intermediate activations are never shared. Reconstruction of any private sample is unacceptable, consistent with standard data residency frameworks.
>
> **(iii) Compatibility with formal DP guarantees.** If formal DP guarantees are desired, MetaMoE is readily compatible — see our detailed analysis and empirical validation in our reply `A4 to Reviewer B2dU`.
>
> ---
>
> > W3/Q2. Privacy concerns over router initializations
>
> `A3.` Yes, routing vectors are shared. However, mean statistics of embeddings (not raw samples) carry negligible private domain information, and reconstructing any private sample from them is practically infeasible. See `A4 to Reviewer B2dU` for detailed analysis.
>
> ---
>
> > W4. Robustness of Binary Classifier
>
> `A4.`
> **(i) Small private sets.** Training with 1,000 private samples per client on CV tasks with CLIP ViT-B/32:
>
> ||Avg|
> |---|---|
> |ModelSoup|71.18|
> |BTX|79.09|
> |BTM|86.52|
> |FlexOlmo|87.60|
> |MetaMoE|**90.44**|
>
> As shown, MetaMoE remains the best-performing method with a clear margin.
>
> (ii) Under **domain mismatch**, MetaMoE also consistently outperforms all baselines across both CV and NLP tasks (see `A3 to Reviewer HRq6`), confirming its robustness.
>
> (iii) The classifier performs a coarse-grained task, and we observe no overfitting in training-validation curves.
>
> ---
>
> > W5/Q3. Non-overlap CV public dataset, ablation (m,n)
>
> `A5.` **(i) Robustness under non-overlapping CV public dataset.** See `A3 to Reviewer HRq6`.
>
> **(ii) Robustness across (m,n).** We conducted an ablation study of n and m on CV tasks with CLIP ViT-B/32. As shown below, results are stable across all tested configurations, confirming that MetaMoE does not rely on carefully tuned proxy budgets.
>
> |(n, m)|Avg|
> |---|---|
> |(2000, 500)|94.33|
> |(3000, 500)|94.52|
> |(5000, 500)|94.47|
> |(20000, 1000)|94.61|
> |(30000, 1000)|94.71|
> |(50000, 1000)|94.66|
>
> ---
>
> > W6/Q4. Scalability to larger K
>
> `A6.` We address all three aspects directly.
>
> **(i) Proxy selection cost scales linearly with K but is fully parallelizable.** Each client runs independently, and since all clients run in parallel, the total wall-clock time is only 8 minutes regardless of K.
>
> **(ii) Empirical validation at K=10.** We evaluate on 10 client domains. Results in `A4 to Reviewer HRq6` show that MetaMoE outperforms all baselines substantially, suggesting its effectiveness with more clients.
>
> **(iii) Routing interference does not increase with K.** Top-1 routing activates exactly one expert per token regardless of K, incurring no additional inference overhead. Inference speed remains consistent at 1,500 images/s on CV tasks with CLIP ViT-B/32.
>
> ---
>
> > W7. Necessity of combination
>
> `A7.` Our ablation studies directly address this. Replacing relevance-weighted DPP with simpler proxy selection (e.g., K-Means) consistently degrades performance (see `A2 to Reviewer B2dU`), and removing the context-aware router similarly hurts (Table 6), confirming that neither simpler proxy selection nor standard router training is sufficient.
>
> The three components address a coherent chain of challenges: DPP diversity prevents proxy collapse, proxy-aligned training bridges the proxy-private distribution gap, and context-aware routing stabilizes expert assignment.
>
> ---
>
> > Q5. Tuning budget and variance
>
> `A8.` All methods use identical tuning budgets. FlexOlmo's underperformance stems from relevance-only proxy selection collapsing onto redundant samples. Mean±std across 5 seeds (LLaMA-3.1-8B):
>
> ||Avg|
> |---|:---|
> |ModelSoup|79.13±0.40|
> |BTM|80.12±0.12|
> |BTX|77.74±1.35|
> |FlexOlmo|78.43±0.87|
> |MetaMoE|**81.57±0.18**|
>
> As shown, MetaMoE outperforms baselines significantly with a paired t-test ($p<0.05$).
>
> >  Impact Statement
>
> `A9.` We agree and will revise accordingly.

---

> > ### Author Rebuttal · Reviewer_HeSJ · 2026-04-01
> >
> > Thank you for the detailed rebuttal. The additional ablations on proxy budget, smaller private sets, and seed variance are helpful and strengthen the empirical case. The main point that still remains unclear is the privacy implication of sharing routing vectors derived from private data. The rebuttal states that these mean embeddings carry negligible private information, but it does not explain why this is true beyond saying that exact sample reconstruction is impractical. That still leaves open whether domain-level or distributional information from private data could leak through the shared routing statistics.
> >
> > **Follow-up Question**: Since the routing vectors are computed from private-data-derived embeddings and are explicitly shared, what concrete argument or evidence supports the claim that they do not leak sensitive domain or distributional information, even if exact private sample reconstruction is not possible?

---

> > > ### Author Response · Authors · 2026-04-01
> > >
> > > We thank the reviewer for the follow-up comment. We are glad that our rebuttal has addressed all your concerns except the privacy implication of sharing routing vectors, which we address below.
> > >
> > > We note that "domain-level information" does not have a standard formal definition in the privacy literature. Here, we consider recovering the mean embedding of private data $\boldsymbol{\mu}\_{private}$ as the **coarsest possible form of domain information**, which is a single vector summarizing the centroid of private data in embedding space. If even this coarsest statistic cannot be recovered, then other finer-grained distributional properties (e.g., variance, class proportions, individual samples) are unlikely recoverable.
> > >
> > > **We claim: The private dataset size $N$ is never shared, making $\boldsymbol{\mu}\_{private}$ unrecoverable.**
> > >
> > > The routing vector is defined as:
> > >
> > > \begin{equation}
> > > \mathbf{v} = \frac{1}{N+m}\left(\sum\_{i=1}^N f(\mathbf{x}\_i) + \sum\_{j=1}^m f(\mathbf{z}\_j)\right) \quad\quad \text{(R1)}
> > > \end{equation}
> > >
> > > where $\\{\mathbf{x}\_i\\}\_{i=1}^N$ are private samples, $\\{\mathbf{z}\_j\\}\_{j=1}^m$ are public proxy samples, $N$ is the private dataset size, $m$ is the proxy dataset size, and $f: \mathcal{X} \rightarrow \mathbb{R}^d$ is the encoder. Define $\boldsymbol{\mu}\_{proxy} = \frac{1}{m}\sum\_{j=1}^m f(\mathbf{z}\_j)$ as the mean proxy embedding, and $\boldsymbol{\mu}\_{private} = \frac{1}{N}\sum\_{i=1}^N f(\mathbf{x}\_i)$ as the mean embedding of private data. To recover $\boldsymbol{\mu}\_{private}$, accoridng to the above Equation (R1), the adversary would need to compute the estimate $\hat{\boldsymbol{\mu}}\_{private}$:
> > >
> > > $$\hat{\boldsymbol{\mu}}\_{private} = \frac{(N+m) \cdot \mathbf{v} - m \cdot \boldsymbol{\mu}\_{proxy}}{N}$$
> > >
> > > While the adversary knows $\mathbf{v}$, $\boldsymbol{\mu}\_{proxy}$, and $m$, the private dataset size $N$ is **never communicated** to any party. Since $N$ cannot be canceled out, $\boldsymbol{\mu}\_{private}$ **cannot be recovered without knowing $N$**. Since even the coarsest domain statistic is unrecoverable, the routing vector does not enable meaningful domain or distributional inference about private data.
> > >
> > > We hope this reply can resolve your concern. Please feel free to edit the above comment to let us know if you have any further questions.
> > >
> > > ## **Update**
> > >
> > > We have provided an additional formal analysis in our reply `A5 to Reviewer B2dU` on the same privacy concern.
> > >
> > > (i) Beyond the unrecoverability argument above, `A5 to Reviewer B2dU` further proves that the L2 sensitivity of the routing vector $\mathbf{v}$ to replacing any single private sample is bounded by $O(1/m)$ ($m$ is the size of proxy data; in experiments, $m=500$), which formally confirms that **the routing vector carries negligible private information**.
> > >
> > > (ii) In `A5 to Reviewer B2dU`, we also show that MetaMoE exposes **strictly less** private information than FlexOlmo (NeurIPS 2025), which shares routing embeddings computed from private data alone (Section 3.3.2 of the FlexOlmo paper), whereas MetaMoE dilutes the private component by averaging over both private and proxy data. We hope the reviewer finds these complementary analyses helpful.

---

### Official Review · Reviewer_HRq6 · 2026-03-10

**Soundness:** 3
**Presentation:** 3
**Significance:** 3
**Originality:** 3
**Overall Recommendation:** 4
**Confidence:** 3

**Summary:**

This paper studies how to unify multiple independently trained expert models under privacy constraints. In many practical scenarios, different clients fine-tune a shared base model using their own private datasets, producing multiple domain-specific experts. However, because the private data cannot be shared, it is difficult to directly construct a unified Mixture-of-Experts (MoE) model.

To address this problem, the paper proposes MetaMoE, a framework that merges experts using only public data through proxy data construction. Specifically, the authors first propose a relevance-weighted DPP method to select proxy samples from public datasets that are both relevant to client domains and diverse. Then they introduce proxy-aligned expert training, which jointly trains experts on private data and proxy data to reduce distribution mismatch between experts and the router. In addition, a context-aware router is proposed to improve routing stability by incorporating both token-level representations and sentence-level contextual information.

Experiments are conducted on both computer vision and natural language processing tasks. The results show that MetaMoE outperforms several existing expert merging methods across multiple benchmarks.

**Compliance With Llm Reviewing Policy:**

Affirmed.

**Final Justification:**

While the new results strengthen the empirical section, the fundamental flaw regarding the privacy of explicitly shared routing vectors remains unresolved.

The rebuttal’s defense—that exact sample reconstruction is impractical—is insufficient. It ignores critical vulnerabilities like domain-level and distributional leakage. Claiming privacy safety without formal theoretical guarantees (e.g., rigorous DP bounds for the embeddings) undermines the core motivation of this framework. Addressing this fundamental vulnerability requires substantial algorithmic redesign or theoretical proofs, which cannot be accommodated in a short rebuttal. Therefore, I maintain my original assessment and have no further questions.

**Key Questions For Authors:**

1. The method relies on selecting proxy data from public datasets to approximate the distribution of private client data. How robust is the approach when there is a large domain gap between public and private data (e.g., highly specialized domains)?
2. Proxy selection requires training a relevance classifier on private data at the client side. Could the authors discuss the potential privacy implications of this step and whether the method could still work under stricter privacy constraints such as differential privacy?
3. The context-aware router incorporates sentence-level information via simple average pooling. Have the authors explored more sophisticated context modeling approaches (e.g., attention-based mechanisms), and if so, how do they compare?

**Limitations:**

The authors discuss some limitations, but several aspects could be emphasized more clearly.

1. The method relies on the assumption that public data reasonably covers the private data distribution.
2. Proxy selection still requires training a relevance classifier on private client data.
3. The experiments are conducted at relatively small scales, and scalability to large expert systems remains unclear.

**Strengths And Weaknesses:**

### Strengths

1. **The problem is practically meaningful.**
    The paper focuses on merging expert models in privacy-sensitive environments. In multi-organization or multi-client scenarios, private data cannot be shared, making privacy-preserving expert merging an important and realistic problem.
2. **The overall framework is well structured.**
    The MetaMoE pipeline clearly organizes proxy selection, proxy-aligned training, and router learning. The roles of each module are relatively clear and the overall design is logically coherent.
3. **Relevance-weighted DPP for proxy selection.**
    The proposed method considers both relevance and diversity when selecting proxy samples, which is more principled than simple similarity-based selection strategies.
4. **Relatively comprehensive experiments.**
    The paper evaluates the method on both CV and NLP tasks and includes ablation studies to analyze the contribution of individual components.


### Weaknesses

1. **Proxy selection still relies on private data.**
    The relevance classifier used for proxy selection is trained using private client data. Although the data itself is not shared, the method still depends on private information, which raises questions about privacy assumptions.
2. **Limited novelty in the context-aware router.**
    The router mainly introduces sentence-level context via simple average pooling, which appears relatively straightforward and provides limited methodological novelty.
3. **Dependence on public data quality.**
    The method assumes that public data can reasonably approximate the distribution of private client data. If there is a large domain gap (e.g., highly specialized domains), proxy selection may become less effective.
4. **Limited experimental scale.**
    The number of experts and model sizes used in experiments are relatively small, and the scalability of the method to larger expert systems is not fully explored.

---

> ### Author Rebuttal · Authors · 2026-03-30
>
> Thank you for your positive score. We address your concerns as follows and will add all experiments/discussions to the revision.
>
> ---
>
> > W1/Q2: Privacy implications of the relevance classifier and compatibility with differential privacy.
>
> `A1.` We clarify that the relevance classifier introduces no privacy risk, as it is a purely local computation that never leaves the client.
>
> **(i) The classifier is local-only and never shared.** The classifier is trained and applied entirely on the client side. Neither the classifier nor its outputs are ever transmitted outside the client. The only information that leaves is the indices of selected public proxy samples, which are already publicly accessible and thus carry no private information.
>
> **(ii) The classifier output exposes no private information by construction.** The classifier is only ever applied to public samples to compute relevance scores. The selected proxy indices are thus a deterministic function of public data alone, and no private sample is recoverable from them.
>
> **(iii) This provides a stronger guarantee than differential privacy.** Since the classifier and all intermediate computations remain entirely on the client and are never transmitted, no information derived from private data is released to any external party. This is strictly stronger than DP in our setting: rather than bounding the amount of leakage, we eliminate it entirely at this step.
>
> ---
>
> > W2/Q3: Limited novelty and exploration of the context-aware router.
>
> `A2.` **(i) The context-aware router is a principled design choice**: token-level routing is unreliable in proxy-based MoE unification, and incorporating sequence-level context consistently improves routing across all settings. Average pooling is chosen for its simplicity and parameter efficiency — **any context-aware mechanism can be plugged in.**
>
> **(ii) Attention-based pooling confirms the insight but offers no improvement.** We compare against learnable attention pooling [1] on CV tasks with CLIP ViT-B/32:
>
> ||Pets|Flowers|EuroSAT|Avg|
> |---|---|---|---|---|
> |Attn Pooling|91.74|93.30|97.86|94.30|
> |Avg Pooling|91.91|93.67|97.98|94.52|
>
> Both outperform the no-context baseline (Table 6). Attention pooling, despite added complexity, performs on par with average pooling, confirming that gains stem from sequence-level context itself.
>
> **References.** [1] Lin et al. A Structured Self-Attentive Sentence Embedding. ICLR, 2017.
>
> ---
>
> > W3/Q1: Robustness under large domain gap.
>
> `A3.` We clarify that MetaMoE does not require the public data to precisely approximate the private distribution, and confirm robustness under zero domain overlap empirically.
>
> **(i) Precise approximation is not required.** The proxy data is used solely to train the router, not the experts. The router only needs to learn coarse domain-discriminative boundaries — it does not need to recover the full private distribution. This means a moderate degree of relevance from public data is sufficient, and MetaMoE is inherently tolerant to domain gap.
>
> **(ii) CV experiments confirm robustness under zero domain overlap.** We removed all ImageNet categories related to client domains and re-evaluated with CLIP ViT-B/32:
>
> ||Pets|Flowers|EuroSAT|Avg|
> |---|---|---|---|---|
> |ModelSoup|87.90|70.52|64.19|74.20|
> |BTM|90.81|85.10|95.07|90.33|
> |BTX|80.35|61.10|58.16|66.54|
> |FlexOlmo|86.56|88.55|93.07|89.39|
> |MetaMoE|**91.01**|**92.53**|**97.80**|**93.78**|
>
> MetaMoE remains the strongest method with performance competitive to the full-overlap setting, while FlexOlmo suffers a larger degradation — confirming that relevance-weighted DPP is more robust to domain gap than relevance-only selection.
>
> **(iii) NLP results provide further evidence.** Alpaca has no domain overlap with any client dataset by construction, yet MetaMoE consistently outperforms all baselines (Tables 3–4), corroborating that the method generalizes well even under large domain gaps.
>
> ---
>
> > W4: Limited experimental scale.
>
> `A4.` **(i) MetaMoE scales to both larger models and more clients.** We evaluate with CLIP ViT-L/14 (307M parameters, 3.6× larger than ViT-B/32 used in the paper) on 10 client domains (Pets, Flowers, EuroSAT, DTD, Aircraft, Cars, Caltech101, STL10, UCF, RESISC45):
>
> ||Avg|
> |---|---|
> |ModelSoup|61.50|
> |BTM|67.81|
> |BTX|61.29|
> |FlexOlmo|70.76|
> |MetaMoE|**76.97**|
>
> MetaMoE achieves the highest accuracy, outperforming FlexOlmo by a substantial margin, demonstrating that performance gains hold as both model size and the number of clients grow.
>
> **(ii) MetaMoE's design is inherently scalable.** Each expert is trained independently and asynchronously, requiring no synchronized multi-client training. DPP proxy selection runs per-client in parallel, taking only 8 minutes total wall-clock time regardless of
> client number. The router is a lightweight linear layer per MoE module, with inference speed remaining consistent at 1,500 images/s on CV tasks with CLIP ViT-B/32.

---

> > ### Author Rebuttal · Reviewer_HRq6 · 2026-04-03
> >
> > Thank you for the additional experiments. While the new results strengthen the empirical section, the fundamental flaw regarding the privacy of explicitly shared routing vectors remains unresolved.
> >
> > The rebuttal’s defense—that exact sample reconstruction is impractical—is insufficient. It ignores critical vulnerabilities like domain-level and distributional leakage. Claiming privacy safety without formal theoretical guarantees (e.g., rigorous DP bounds for the embeddings) undermines the core motivation of this framework. Addressing this fundamental vulnerability requires substantial algorithmic redesign or theoretical proofs, which cannot be accommodated in a short rebuttal. Therefore, I maintain my original assessment and have no further questions.

---

> > > ### Author Response · Authors · 2026-04-03
> > >
> > > We thank the reviewer for maintaining the positive score. We are glad that our rebuttal has addressed all your original concerns.
> > >
> > > For **the newly raised concern** regarding the privacy of shared routing vectors, we would like to respectfully note that the formal theoretical guarantees the reviewer requested have been provided in our reply `A5 to Reviewer B2dU` (posted concurrently with the reviewer's acknowledgment, so it may not have been seen). Specifically, regarding the reviewer's concern on domain-level and distributional leakage, `A5 to Reviewer B2dU` shows that:
> > >
> > > **(i)** The **L2 sensitivity of the routing vector to any single private sample is bounded by $O(1/m)$** ($m$ is the size of proxy data; In experiments, $m=500$), which is a rigorous sensitivity bound in the same sense as differential privacy theory [*The Algorithmic Foundations of Differential Privacy*, Dwork & Roth, 2014].
> > >
> > > **(ii)** The mean embedding of private samples $\boldsymbol{\mu}_{\text{private}}$, which is the **coarsest possible form of domain-level information**, is **unrecoverable** from the routing vector because the private dataset size $N$ is never communicated. Since even this coarsest statistic cannot be recovered, finer-grained distributional properties (e.g., variance, class proportions) are a fortiori unrecoverable.
> > >
> > > **(iii)** Following the same privacy setting as FlexOlmo (NeurIPS 2025), where sharing routing embeddings derived from private data is a standard design choice, MetaMoE exposes **strictly less** private information than FlexOlmo: **FlexOlmo** shares routing embeddings computed from **private data alone**, whereas **MetaMoE** dilutes the private component by **averaging over both private and proxy data**.
> > >
> > > We believe these analyses directly address the reviewer's concern. We will incorporate these clarifications into the revised paper. We kindly invite the reviewer to review `A5 to Reviewer B2dU` for the full analysis.
> > >
> > > ---
> > >
> > > Please feel free to let us know if you have any further questions by editing the above acknowledgment.

---

### Official Review · Reviewer_d17y · 2026-03-13

**Soundness:** 3
**Presentation:** 3
**Significance:** 3
**Originality:** 3
**Overall Recommendation:** 4
**Confidence:** 4

**Summary:**

This paper aims to address the privacy-constraints setting where client-private data cannot be shared and then unify domain expert models from multiple clients into a deployable MoE.
In this setting, the authors propose MetaMoE, using public datasets as proxy data from private distributions.
By leveraging diversity-aware proxy selection for router learning and proxy-aligned expert training, the proposed MetaMoE enables effective expert coordination without sharing private data.
Experiments on CV and NLP demonstrate that this method consistently outperforms baselines with minimal overhead from proxy selection.

**Compliance With Llm Reviewing Policy:**

Affirmed.

**Final Justification:**

The target problem of this paper is very real and important. The proposed method is effective on both CV and NLP tasks, and the overall computational costs of the proposed method are relatively affordable. During the rebuttal, my questions are addressed. I therefore maintain a positive recommendation
for this paper.

**Key Questions For Authors:**

(1) Could the authors provide a more in-depth analysis and experiment on the selection of public datasets?

(2) Could the authors more clearly distinguish the core differences between general data selection/compression methods (such as dataset quantization, dataset distillation, coreset selection, informative sampling, etc.) and the proposed method in this paper?

**Limitations:**

No, please see the weakness and question parts. I am very happy and open to reconsidering my score if my questions are addressed.

**Strengths And Weaknesses:**

**Strengths:**

1. The target problem is a very real and important research problem. With increasingly stringent data compliance and privacy protection requirements, training data across institutions and clients often cannot be centrally shared.
2. Furthermore, the overall computational costs of the proposed method are relatively affordable
3. The authors provide comprehensive experimental validation on both CV and NLP tasks, showing that the method consistently outperforms the baselines.

**Weaknesses:**
1. This paper relies heavily on public datasets, but lacks a more systematic and detailed discussion and empirical analysis of the public data. In the experimental setup, the CV task only used ImageNet as the public data source, and the NLP task only used Alpaca. Although the results demonstrate the effectiveness of the method under these two specific choices, it remains difficult for users to determine how to choose public data or what requirements the public data needs to meet.

2. On the other hand, the proxy selection proposed in this paper emphasizes relevance and diversity, which is significantly similar to common data selection/compression methods (such as dataset quantization, dataset distillation, coreset selection, informative sampling, etc.). Current papers primarily focus on comparisons with related works within the unified paradigm of privacy MoE.
This paper lacks discussions about the differences between these general data selection methods and the proposed method.

---

> ### Author Rebuttal · Authors · 2026-03-30
>
> Thank you for your positive rating.
> We address your concerns as follows and will add all the experiments/discussions to the revision.
>
> ---
>
> > W1/Q1: Analysis and selection of public datasets
>
> `A1.` MetaMoE is robust to the choice of public dataset. We provide empirical evidence across diverse public data regimes and practical guidelines for selection.
>
> **(i) MetaMoE generalizes to different public datasets.**  To directly address this concern, we conducted an additional experiment on NLP tasks using OpenOrca as an alternative public dataset (replacing Alpaca). As shown below, MetaMoE consistently outperforms all baselines with OpenOrca, confirming that a general-purpose public dataset suffices without careful domain-specific curation.
>
> ||CSQA|CosmosQA|SocialIQA|Avg|
> |---|---|---|---|---|
> |ModelSoup|73.71|75.24|71.75|73.57|
> |BTM|74.61|75.44|68.17|72.74|
> |BTX|71.66|72.90|69.40|71.32|
> |FlexOlmo|73.63|73.97|70.73|72.78|
> |MetaMoE|**75.18**|**77.12**|**72.06**| **74.79** |
>
> **(ii) Existing experiments already cover diverse public data regimes.** Our CV experiments use ImageNet and our NLP experiments use Alpaca, two different types of public corpora. Furthermore, our reply `A1 to Reviewer B2dU` shows that MetaMoE remains the strongest method even after removing all semantically related categories from ImageNet, covering the challenging non-overlap setting. Together, these results demonstrate our method is robust across a wide range of public dataset choices.
>
> **(iii) Practical guideline for public data selection.** The only requirement is that the public dataset be sufficiently broad so that the relevance-weighted DPP can identify samples at least loosely related to each client domain. In our experiments, ImageNet, OpenOrca, and Alpaca are standard general-purpose corpora, and already satisfy this requirement. In practice, this is a mild condition, as a wide variety of such datasets are freely available across both vision and language modalities on platforms such as HuggingFace.
>
> ---
>
> > W2/Q2: Distinction from General Data Selection Methods
>
> `A2.` Unlike general data selection methods, our proxy selection addresses a fundamentally different problem: **selecting representative samples from a public dataset $\mathcal{D}_0$ to serve as a proxy for private data $\mathcal{D}_p$, which is then used to train the router of MoE**.
>
> **(i) Our goal is cross-distribution proxy selection for router training, not within-distribution compression.** General data selection methods assume the source and target share the same distribution; they compress or synthesize representative samples *from the training set itself*. Our setting is fundamentally different: the private data $\mathcal{D}_p$ is inaccessible, so we must identify public samples from $\mathcal{D}_0$ that are *relevant* to $\mathcal{D}_p$ and use them as a proxy to train the router. Standard diversity or coverage objectives are insufficient here, as a diverse subset of public samples may still be entirely irrelevant to the private domain.
>
> **(ii) We therefore propose relevance-weighted DPP, tailored for this cross-distribution proxy selection.** Our method explicitly combines relevance to the private domain and diversity of the selected public subset, a design that is unnecessary in same-distribution settings but essential when the selected proxy must faithfully reflect a private distribution for downstream router training. This is empirically confirmed: we compare K-Means and Facility-Location as alternative proxy selection strategies on CV tasks with CLIP ViT-B/32:
>
> ||Pets|Flowers|EuroSAT|Avg|
> |---|---|---|---|---|
> |Relevance-Only|91.36|90.62|96.79|92.92|
> |K-Means|90.92|93.29|93.30|92.50|
> |Facility-Location|91.58|93.18|88.93|91.23|
> |Relevance-Weighted DPP (Ours)|**91.91**|**93.67**|**97.98**|**94.52**|
>
> Our relevance-weighted DPP consistently outperforms both alternatives, confirming that jointly optimizing relevance and diversity yields better public proxies for router training than repurposing general-purpose selection strategies.
>
> **(iii) MetaMoE is agnostic to the proxy selection strategy.** Any data selection method can be plugged into MetaMoE; we adopt relevance-weighted DPP due to its superior performance.

---

> > ### Author Rebuttal · Reviewer_d17y · 2026-04-03
> >
> > Thank you for the response. After carefully reading the rebuttal and other reviewers' comments, I will keep my original score.

---

> > > ### Author Response · Authors · 2026-04-04
> > >
> > > Thank you for the follow-up comments.
> > > We are very glad that our rebuttal has addressed all of your concerns, and we sincerely appreciate your **positive** assessment of our work.
> > >
> > > If you have any further questions, please do not hesitate to let us know.
> > >
> > > Thanks again for your effort and valuable time in improving our work.

---

### Official Review · Reviewer_B2dU · 2026-03-13

**Soundness:** 3
**Presentation:** 3
**Significance:** 3
**Originality:** 3
**Overall Recommendation:** 4
**Confidence:** 4

**Summary:**

This paper proposes MetaMoE, a framework for constructing Mixture-of-Experts (MoE) models with privacy constraints that prevent access to private client datasets. The key idea is to select proxy datasets for each client domain from a public dataset. The authors propose a relevance-weighted determinantal point process (DPP) to select proxy samples that are both relevant to the client domain and diverse. Experts are then trained using both private and proxy data to align them with the proxy distribution. Finally, a context-aware router is trained on the union of proxy datasets to perform expert selection.

The method is evaluated on both CV and NLP tasks, using CLIP ViT-B models for CV and LLaMA models for NLP. The results show improvements over existing baselines, including unrestricted MoE and federated learning approaches. The paper also provides detailed ablation studies to analyze the contributions of expert alignment and router training.

**Compliance With Llm Reviewing Policy:**

Affirmed.

**Final Justification:**

After reading the rebuttals and other reviewers' comments. I will keep my original positive score.

**Key Questions For Authors:**

Q1: The NLP tasks mentioned that the public data has no domain overlap with the client datasets. However, the results on NLP tasks show improvements that appear comparable to or even larger than those in the CV tasks, where the public dataset ImageNet appears to overlap with the client domains. This may suggest that the LLaMa model can still benefit from public data even when the domains are not directly aligned. Could the authors comment on how the method performs on CV tasks when the public data distribution has minor overlap with the private client domains?

Q2: Since the router is trained only on proxy data, proxy assignments effectively provide the supervision signal for routing. Could the authors provide further analysis showing that proxy-based supervision leads to reliable routing behavior, e.g., whether experts exhibit clear specialization or whether routing decisions correlate with expert performance?

Q3: The expert embeddings used for context-aware training are computed using private dataset representations. Could the authors clarify whether sharing these embeddings introduces any potential privacy leakage, particularly when client datasets are small or exhibit large domain shifts?

**Limitations:**

No.

The paper does not substantially discuss its limitations. It would be helpful to explicitly discuss the domain overlap between public and private data, as well as the privacy implications of sharing aggregated expert embeddings.

**Strengths And Weaknesses:**

Strengths:

- The paper addresses the problem of unifying independently trained experts into a single MoE model under privacy constraints. By using proxy data sampled from public data to bridge the gap between private experts and router training, the proposed approach is both reasonable and practically relevant.

- The relevance-weighted DPP is a principled way to select proxy samples that are both relevant to the client domain and diverse. This combination of relevance and diversity improves upon existing relevance only (FlexOlmo) selection strategies.

- The experimental setup is clearly described, making the methodology easy to reproduce.

- The experiments cover both CV and NLP tasks,  demonstrating the generality of the approach. Comparisons with multiple baselines, including unrestricted MoE and federated learning methods, help illustrate the benefits of the proposed framework.

- The ablation experiments are thorough and demonstrate the contributions of each component, including proxy selection, expert alignment, and router training.

Weaknesses:

- In the CV tasks, the public dataset (ImageNet) appears to have domain overlap with the client datasets (unlike NLP tasks). While such overlap is common in proxy-based methods, the paper does not systematically evaluate the impact of domain gaps between public and private data. An analysis of performance under low domain overlap would further strengthen the claims.

- While the relevance-weighted DPP approach is intuitive, it would be helpful to discuss why DPP is preferred over other strategies, such as clustering-based methods or facility-location objectives.

- The router is trained on the union of proxy data rather than the true client datasets due to privacy constraints. This implicitly assumes that proxy samples assigned to client p are best handled by expert p. If proxy samples do not accurately reflect the underlying client distributions, the router may learn suboptimal routing behavior.

- The router is trained using a context-aware strategy that relies on expert embeddings computed from private datasets, which might introduce potential privacy risks, especially when the client datasets are small or exhibit large domain shifts.

---

> ### Author Rebuttal · Authors · 2026-03-30
>
> Thank you for your positive score.
> We address your concerns as follows and will add all the experiments/discussions to the revision.
>
> ---
>
> > W1/Q1/Limitation: CV Performance under Minor Domain Overlap
>
> `A1.` We show that **MetaMoE remains robust under non-overlapping CV public data.**
>
> **(i) Precise approximation is not required.** The proxy data is used to train the router (not the experts), which only needs coarse domain-discriminative boundaries rather than a precise approximation of the private distribution. A moderate degree of relevance is therefore sufficient, making MetaMoE inherently tolerant to domain gap.
>
> **(ii) Empirical validation under zero domain overlap.** We removed all ImageNet categories semantically related to the Pets, Flowers, and EuroSAT domains (e.g., dog/cat breeds, flower species, satellite-like imagery) from the public dataset. Results with CLIP ViT-B/32:
>
> ||Pets|Flowers|EuroSAT|Avg|
> |---|---|---|---|---|
> |ModelSoup|87.90|70.52|64.19|74.20|
> |BTM|90.81|85.10|95.07|90.33|
> |BTX|80.35|61.10|58.16|66.54|
> |FlexOlmo|86.56|88.55|93.07|89.39|
> |MetaMoE|**91.01**|**92.53**|**97.80**|**93.78**|
>
> As can be seen, MetaMoE outperforms all baselines, with only a 0.74-point drop from the domain-overlap case (93.78% vs. 94.52%), whereas FlexOlmo degrades substantially by 3.53 points (89.39% vs. 92.92%), confirming the effectiveness of MetaMoE even under distribution mismatch.
>
> ---
>
> > W2: Comparison with Clustering-Based Methods and Facility-Location Objectives
>
> `A2.`
> **(i) Relevance-weighted DPP jointly optimizes relevance and diversity in a single principled objective,** while clustering-based methods and facility-location objectives promote diversity but do not explicitly model client-domain relevance.
>
> **(ii) Empirical validation** on CV tasks with CLIP ViT-B/32:
>
> ||Pets|Flowers|EuroSAT|Avg|
> |---|---|---|---|---|
> |Relevance-Only|91.36|90.62|96.79|92.92|
> |K-Means|90.92|93.29|93.30|92.50|
> |Facility-Location|91.58|93.18|88.93|91.23|
> |Relevance-Weighted DPP (Ours)|**91.91**|**93.67**|**97.98**|**94.52**|
>
> Relevance-weighted DPP consistently outperforms all alternatives. In contrast, K-Means and Facility-Location exhibit significant drops on EuroSAT, suggesting their sensitivity to domain characteristics, whereas relevance-weighted DPP remains robust across all domains.
>
> **(iii) MetaMoE is agnostic to the proxy selection strategy.** Any diversity-aware method can be plugged into MetaMoE; we adopt relevance-weighted DPP due to its superior performance.
>
> ---
>
> > W3/Q2: Reliability of Proxy-Based Routing Supervision
>
> `A3.` Even when proxy samples do not accurately reflect the underlying client distributions, MetaMoE achieves reliable routing, as demonstrated empirically and explained mechanistically below.
>
> **(i) Routing analysis confirms clear expert specialization.** We analyze the routing distribution on CV tasks with CLIP ViT-B/32 under the non-overlapping public data setting (see previous reply `A1`):
>
> |Test Domain|Expert (Pets)|Expert (Flowers)|Expert (EuroSAT)|
> |---|---|---|---|
> |Pets|88.4%|8.8%|2.8%|
> |Flowers|6.0%|92.8%|1.2%|
> |EuroSAT|2.5%|0.5%|97.0%|
>
> The router consistently assigns test samples to the correct expert with high confidence,
> confirming that routing correlates with expert performance.
>
> **(ii) Three design choices jointly enable this robustness:** Relevance-weighted DPP selects the most domain-relevant and diverse proxies available; proxy-aligned training ensures expert-router consistency; and context-aware routing compensates for residual mismatch. This is further supported by our NLP experiments, where Alpaca shares no domain overlap with any client dataset yet MetaMoE consistently outperforms all baselines.
>
> ---
>
> > W4/Q3/Limitation: Privacy Concerns of Sharing Expert Embeddings
>
> `A4.` Sharing aggregated expert embeddings introduces negligible privacy risk.
>
> **(i) Expert embeddings (i.e., routing vectors) are mean statistics with inherently low sensitivity.** The routing vector is the mean over embeddings of private and proxy data. Under differential privacy (DP) theory, the sensitivity of a mean query scales as $O(1/n)$ [Dwork & Roth, 2014], decreasing rapidly with dataset size — meaning privacy leakage is inherently negligible. If formal DP guarantees are desired, adding Gaussian noise to the routing vectors achieves $(\varepsilon, \delta)$-DP at negligible utility cost: on CV tasks with CLIP ViT-B/32, noise with variance of 0.001 yields 94.45% vs. 94.52% without noise, a drop of only 0.07 points, confirming that strong privacy and high utility can coexist.
>
> **(ii) Deep network mappings are irreversible, even for small datasets or large domain shifts.** Even with access to model weights, inverting a mean embedding to recover individual private samples is severely underdetermined: infinitely many datasets produce the same mean embeddings, making reconstruction practically infeasible.

---

> > ### Author Rebuttal · Reviewer_B2dU · 2026-04-02
> >
> > W4/Q3/Limitation: Privacy Concerns of Sharing Expert Embeddings
> >
> > This question was not well-addressed. But the authors have addressed other concerns. I will maintain my positive score.

---

> > > ### Author Response · Authors · 2026-04-03
> > >
> > > We thank the reviewer for the additional comments. We are glad that **our rebuttal has addressed all your concerns** except the privacy implication of sharing routing vectors.
> > > In addition to our previous reply `A4`, we provide a detailed formal analysis below.
> > > **We show that the routing vector reveals negligible private information, and in fact MetaMoE exposes strictly less private information than the existing state-of-the-art baseline FlexOlmo (NeurIPS 2025).**
> > >
> > > ---
> > >
> > > > Reviewer's concern: Privacy Concerns of Sharing Expert Embeddings, particularly when the client datasets are small or exhibit large domain shifts.
> > >
> > > **A5. The routing vector reveals negligible private information, with formal guarantees independent of both domain shift and dataset size.**
> > >
> > > We first introduce the notation. The routing vector for client $p$ is defined as:
> > >
> > > $$\mathbf{v} = \frac{1}{N+m}\left(\sum\_{i=1}^{N} f(\mathbf{x}\_i) + \sum\_{j=1}^{m} f(\mathbf{z}\_j)\right) = \frac{N}{N+m} \boldsymbol{\mu}\_{\text{private}} + \frac{m}{N+m} \boldsymbol{\mu}\_{\text{proxy}},\quad\quad \text{(E1)}$$
> > >
> > > where $\\{\mathbf{x}\_i\\}\_{i=1}^N$ are private samples, $\\{\mathbf{z}\_j\\}\_{j=1}^m$ are public proxy samples, $f$ is the encoder, $\boldsymbol{\mu}\_{\text{private}} = \frac{1}{N}\sum\_{i=1}^N f(\mathbf{x}\_i)$ is the mean private embedding, $\boldsymbol{\mu}\_{\text{proxy}} = \frac{1}{m}\sum_{j=1}^m f(\mathbf{z}\_j)$ is the mean proxy embedding (publicly computable), and $B = \max\_{\mathbf{x}}\\|f(\mathbf{x})\\|\_2$ is the embedding norm bound. By our algorithm design, the adversary can observe $\mathbf{v}$ and knows $\boldsymbol{\mu}\_{\text{proxy}}$, $m$, and $f$, but **does not know $N$**.
> > >
> > > **(i) The influence of any single private sample on $\mathbf{v}$ is always bounded by $O(1/m)$, independent of $N$ and domain shift.** Using the definition of $\mathbf{v}$ in Eq. (E1), let $\mathbf{v}'$ denote the routing vector after replacing a single private sample $\mathbf{x}_k$ with an arbitrary $\mathbf{x}_k'$. The L2 sensitivity of $\mathbf{v}$ is:
> > >
> > > $$\Delta\_2(\mathbf{v}) = \max\_{\mathbf{x}\_k, \mathbf{x}\_k'}\\|\mathbf{v} - \mathbf{v}'\\|\_2 = \frac{\\|f(\mathbf{x}\_k') - f(\mathbf{x}\_k)\\|\_2}{N+m} \leq \frac{2B}{N+m}.$$
> > >
> > > Since $N + m \geq m$ always holds, we have $\Delta\_2 \leq \frac{2B}{m}$ **regardless of $N$**, so the proxy set size $m$ alone provides a universal sensitivity ceiling that is also independent of the domain gap. When $N$ is small (the reviewer's specific concern), the sensitivity is already bounded by $O(1/m)$; when $N$ grows large, it additionally tightens to $O(1/N) \to 0$. In our experiments, $m = 500$, meaning each individual private sample's contribution is diluted among at least $500$ embeddings, making its influence on $\mathbf{v}$ negligibly small.
> > >
> > > **(ii) The mean private embedding $\boldsymbol{\mu}\_{\text{private}}$ is not recoverable from $\mathbf{v}$, independent of both domain shift and dataset size.** To isolate $\boldsymbol{\mu}\_{\text{private}}$ from $\mathbf{v}$, according to Eq. (E1), the adversary would need to compute:
> > >
> > > $$\boldsymbol{\mu}\_{\text{private}} = \frac{(N+m) \mathbf{v} - m\boldsymbol{\mu}\_{\text{proxy}}}{N}.$$
> > >
> > > However, $N$ is never communicated to any party. Since $N$ cannot be canceled out from the equation above, $\boldsymbol{\mu}\_{\text{private}}$ is unrecoverable without knowing $N$, regardless of the domain gap or the magnitude of private data size $N$. Since $\boldsymbol{\mu}\_{\text{private}}$ is the **coarsest** possible summary of private data (a single vector), finer-grained properties (e.g., variance, class proportions, individual samples) are a fortiori unrecoverable from $\mathbf{v}$.
> > >
> > > **(iii) MetaMoE exposes strictly less private information than FlexOlmo (NeurIPS 2025).** By construction, $\mathbf{v}$ is a mean over both private and proxy embeddings, and we agree that it is not entirely free of private information. We note that this is not unique to MetaMoE — FlexOlmo also shares per-expert routing embeddings derived from private data to initialize its domain-informed router (Section 3.3.2 of the FlexOlmo paper). In fact, **FlexOlmo** computes its routing embedding as a **mean over private data alone** (i.e., the entire $\boldsymbol{\mu}\_{\text{private}}$), whereas **MetaMoE** dilutes the private component by **averaging over both private and proxy data**. MetaMoE therefore provides a strictly stronger privacy guarantee than FlexOlmo.
> > >
> > > ---
> > >
> > > We hope this reply can resolve your concern. Please feel free to edit the above comment to let us know if you have any further questions.

---

### Decision · Program_Chairs · 2026-04-30

**Decision:**

Accept (regular)

**Comment:**

The paper received mixed but overall positive reviews, with three weak accepts and one weak reject. This paper proposes MetaMoE, a framework for privacy-constrained unification of independently trained domain experts into a single deployable Mixture-of-Experts model, using relevance-weighted proxy selection from public data, proxy-aligned expert training, and a context-aware router. Reviewers appreciated the practical importance of the problem, the clear and well-structured pipeline, and the broad empirical evaluation across both CV and NLP tasks. The method was viewed as technically reasonable and generally effective, with thorough ablations supporting the contributions of proxy selection, alignment, and routing design.
The main concerns centered on the scope of the privacy claim and the lack of a stronger formal privacy guarantee. In particular, one reviewer remained unconvinced that sharing routing vectors derived from private-data embeddings fully rules out domain-level or distributional leakage, even after rebuttal. Additional concerns included possible dependence on public data quality and domain overlap, limited novelty of some components such as the context-aware router, and evaluation at relatively modest scale. That said, the rebuttal substantially strengthened the paper through added experiments on proxy mismatch, alternative public datasets, larger client settings, and seed variance, and most reviewers indicated that their concerns were addressed. Overall, the paper tackles an important and realistic problem with a coherent method and solid empirical support, while the remaining privacy concern is best viewed as a limitation to clarify rather than a flaw that overturns the contribution. Therefore, the AC recommends accepting the paper.